# Efficient DP-SGD for LLMs with Randomized Clipping

**Enayat Ullah** [1]  **Sai Aparna Aketi** [* 1]  **Devansh Gupta** [* 2]  **Huanyu Zhang** [1]  **Meisam Razaviyayn** [2]

## Abstract

Large language models (LLMs) are trained on vast datasets that may contain sensitive information. Differential privacy (DP), the de facto standard for formal privacy guarantees, provides a principled framework for training LLMs with provable privacy protection. However, state-of-the-art DP training implementations rely on *fast gradient clipping* techniques with memory overhead $O(B \min\{T^2, d^2\})$, where $B$ is the batch size, $T$ is the sequence length, and $d$ is the layer width. This becomes prohibitive as both model width and context length grow. We propose DP-SGD-RC, a novel variant of DP-SGD with *randomized clipping* that reduces memory and compute overhead. DP-SGD-RC leverages *stochastic trace estimation* methods, specifically *Hutchinson's estimator* (Hutchinson, 1989) and its improved variant, Hutch$^{++}$(Meyer et al., 2021), to reduce the memory footprint of per-sample gradient norm estimation. We provide a tight privacy analysis showing that DP-SGD-RC achieves noise multipliers competitive with deterministic clipping. Experiments fine-tuning Llama 3.2 1B on long-context benchmarks spanning classification, question answering, and summarization tasks demonstrate that DP-SGD-RC matches baseline utility while significantly reducing memory and compute.

## 1. Introduction

Differential Privacy (DP) is the de-facto standard notion of data privacy for statistical and machine learning tasks (Dwork et al., 2006). DP provides a quantitative protection against identification of participation of a datum (usually an individual's data) in the data analysis or machine learning task. Consequently, DP has seen rapid growth in development and deployment including that in US Census (Abowd, 2018), Google (Erlingsson et al., 2014), Apple (Apple Differential Privacy Team, 2017) etc.

In the machine learning context, the canonical DP training algorithm is *Differentially Private Stochastic Gradient Descent (DP-SGD)*, (Bassily et al., 2014; Abadi et al., 2016). It is a simple modification to its non-private counter-part, SGD, with two changes: (a). per-sample gradient clipping, (b). addition of calibrated noise to the mini-batch gradient. These changes, though simple, introduce additional complexities. A primary challenge emerges from the need to do per-sample gradient clipping. Naively, it requires instantiating per-sample gradients which leads to prohibitively large, $O(BLd^2)$, memory overhead for batch size $B$, layers $L$ and width $d$ neural nets. This severely limits the application of DP-SGD to modern large scale settings.

Subsequent works proposed *Fast Gradient Clipping* (FGC) (Lee & Kifer, 2021) and *Ghost Clipping* (GC) (Li et al., 2021) which improved the memory overhead to $O(Bd^2)$ and $O(B)$ (for linear-like layers) respectively. The key idea is to implement per sample gradient clipping as per-sample gradient norm estimation followed by loss re-scaling. Further, for standard neural net modules (such as linear layers and non-sequential inputs), per sample gradient norm estimation can be done without instantiation the gradient. This brings the computational aspects of DP-SGD closer to that of non-private training.

Unfortunately, the above improvement isn't realized in all settings. In particular, for sequential inputs (such as text) with context size $T$, the state-of-the-art memory complexity overhead, from the best of FGC and GC, is $O(B \min\{d^2, T^2\})$. This is prohibitive even for moderate $T$. This is especially relevant as LLMs have become a dominant part of our daily lives. The state-of-the-art LLMs are trained on potentially sensitive text data and require privacy-preserving techniques to prevent leakage of private information. Further, the context size, $T$, in frontier LLMs has grown to $O(100K)$ to realize agentic capabilities (Grattafiori et al., 2024).

In this work, we study differentially private training of models with sequential (text) data, such as LLMs, with the goal of designing more efficient yet performant algorithms.

---

[*]Equal contribution   [1]Meta Platforms Inc.   [2]University of Southern California. Correspondence to: Enayat Ullah <enayat@meta.com>.

*Proceedings of the 43$^{rd}$ International Conference on Machine Learning*, Seoul, South Korea. PMLR 306, 2026. Copyright 2026 by the author(s).

*Table 1.* Comparison of memory and compute overheads for variants of DP-SGD implementations as compared to non-private implementation of SGD. Herein, $T$: Context length, $B$: Batch size, $(p, d)$: layer shape. DP-SGD-RC uses Hutchinson's estimator with projection dimension $k$ to compute the per-sample gradient norm. For simplicity, we assume $k \ll d < p$. More details can be found in Appendix E.

|  | DP-SGD (Naive) | FGC | GC | **DP-SGD-RC (ours)** |
|---|---|---|---|---|
| Compute | $\mathcal{O}(BTpd)$ | $\mathcal{O}(BTpd)$ | $\mathcal{O}(BT^2(p+d))$ | $\mathcal{O}(BTk(p+d))$ |
| Memory | $BpLd$ | $Bpd$ | $2BT^2$ | $BkT + pk$ |

## 1.1. Contributions

**DP-SGD with Randomized Clipping:** We propose DP-SGD-RC, a variant of DP-SGD with randomized clipping. This improves the memory and compute complexity of SOTA implementations of DP-SGD for sequential data as detailed in Table. 1. In particular, for sequence length $T$, linear (like) layer of shape $(p, d)$, batch size $B$, the memory overhead essentially improves from *quadratic*, $B \min (dp, T^2)$, to *linear*, $BkT + kp$, yielding no asymptotic overhead on non-private training, where $k$ is the projection dimension used in DP-SGD-RC, and is small ($k \approx 32$ in our experiments). The key to this improvement stems from framing the per-sample gradient norm estimation as *stochastic trace estimation*, and using off-the-shelf estimators such as Hutchinson's estimator (**Hutch**) (Hutchinson, 1989) and its improved variant **Hutch**[++] (Meyer et al., 2021).

**Privacy analysis:** We provide a novel privacy analysis for DP-SGD-RC with **Hutch** and **Hutch**[++] used as norm estimation routines. In contrast to a fixed noise scale in DP-SGD, the randomized clipping variant averages the single-step Gaussian privacy kernels over a random scale induced by the norm estimation. We show that in this case, the argument essentially reduces to computing an *envelope* CDF of a convex combination of chi-squared random variables. We give explicit and numerical estimates of the envelope CDF leveraging results and tools from *stochastic orders* and *majorization*. We also provide an efficient accounting algorithm based on PRV (Gopi et al., 2021) with the symbolic description of *envelope* CDF as input. For practical setups, we find that the resulting noise multipliers are close to those from *deterministic* clipping.

**Empirical Evaluation:** Experiments with Llama 3.2 1B across diverse long-context tasks (classification, summarization, and question answering) demonstrate that our method maintains baseline-level performance while achieving up to 40% memory reduction and $2\times$ compute savings for the largest linear layer.

## 2. Related Work

Early work on differentially private machine learning introduced *objective and output perturbation* (Chaudhuri et al., 2011; Kifer et al., 2012), followed by improved guarantees

via DP-SGD (Bassily et al., 2014). DP-SGD was further extended to non-convex setups and its privacy analysis was refined by (Abadi et al., 2016), enabling its practical use in deep learning. While alternative DP optimization methods exist (Asi et al., 2021; Feldman et al., 2020; Arora et al., 2023; Menart et al., 2024; Choquette-Choo et al., 2022; Tang et al., 2024), they often rely on restrictive assumptions or scale poorly. Consequently, DP-SGD has become the de-facto standard, combining strong theory with practical effectiveness and production-grade libraries (Yousefpour et al., 2021).

A key limitation of DP-SGD is the cost of per-sample gradient clipping, which scales poorly with model and batch size. Goodfellow (2015) showed that per-sample gradients in feedforward networks admit efficient layer-wise norm computation. This motivated Fast Gradient Clipping (FGC) (Lee & Kifer, 2021), which enforces clipping by rescaling per-sample losses without materializing full per-sample gradients, at the cost of two backward passes. However, FGC still realizes per-sample gradients one layer at a time, so memory depends on the largest layer. For language models, FGC has memory overhead $O(BTd^2)$, increasing with context length $T$ and width $d$. Li et al. (2021) proposed Ghost Clipping (GC), improving norm-computation overhead for *linear-like* layers to $O(BT^2)$, with large benefits for LLMs with short contexts. Bu et al. (2023) further improved this via *Mixed Ghost Clipping* and *Book-keeping* – Mixed Ghost Clipping chooses between FGC and GC to minimize memory overhead, and Book-keeping reduces the compute cost of the two backward passes. These methods leave DP-SGD unchanged while providing scalable implementations. Orthogonal approaches improve utility by adapting or correcting clipping thresholds (Andrew et al., 2021; Zhang et al., 2024) or restricting private updates to low-rank subspaces (Yu et al., 2022) – these can be combined with FGC and GC.

**Comparison with DP-SGD-JL (Bu et al., 2021):** The most related work to ours is (Bu et al., 2021) which proposes the use of Johnson-Lindenstrauss (JL) projections to compute approximate per-sample gradient norms for DP-SGD. For comparison, in the following, we limit to non-sequential data ($T = 1$) with one linear layer of dimensions $(p, d)$. DP-SGD-JL projects the flattened gradient $\mathbb{R}^{dp} \to \mathbb{R}^k$. In contrast, we sketch the *factored matrix form* of the gradient $(A^\top G)$ per-layer projecting as $\mathbb{R}^{p \times d} \to \mathbb{R}^{k \times d}$. In the

special case of $d = 1$, our method with Hutchinson's as the norm estimator *essentially* reduces to DP-SGD-JL. Further, we build on the privacy analysis technique of DP-SGD-JL with trade-off functions, however, our analysis and result are significantly more complex as we generalize it to $d > 1$. Finally, the two methods differ in their implementations aspects. DP-SGD-JL employs the Jacobian-Vector-Product (JVP) mode of automatic differentiation to compute projected per-sample gradients for norm estimation. While JVP reduces memory footprint by avoiding materialization of full per-sample gradients, DP-SGD-JL requires $k + 1$ forward passes and one backward pass for a projection dimension of $k$. In contrast, our implementation leverages forward and backward *hooks* per layer, similar to Fast Gradient Clipping, requiring only one forward pass and two backward passes.

**Stochastic Trace Estimation:** The problem deals with the design of algorithms to estimate trace$(A)$ without explicit access to $A$. Popular computational models include matrix-vector (Meyer et al., 2021) and Kronecker matrix-vector (Meyer & Avron, 2023) access. In our setting, we apply standard estimators, Hutch and Hutch$^{++}$ and discuss them in more detail in Section 4.

## 3. Preliminaries

In this section, we present the relevant preliminaries, starting with the definition of differential privacy.

**Definition 3.1** (Differential Privacy (Dwork et al., 2014)). Let $\varepsilon \geq 0$, $\delta \in [0, 1)$. A randomized algorithm $M$ is $(\varepsilon, \delta)$-differentially private (DP) if for all pairs of neighbouring data sets $\mathcal{D}, \mathcal{D}'$ and any measurable set $O$,

$$\mathbb{P}(M(\mathcal{D}) \in O) \leq e^\varepsilon \mathbb{P}(M(\mathcal{D}') \in O) + \delta. \qquad (1)$$

**Tradeoff Function and $f$-DP.** We introduce quantities to discuss the modern $f$-DP definition. Let $P$ and $Q$ be the output distributions of a mechanism on neighboring datasets $\mathcal{D}$ and $\mathcal{D}'$. The **privacy loss variable** is defined as:

$$L(x) = \log \left( \frac{p(x)}{q(x)} \right).$$

This quantity captures how much information a single output $x$ reveals about which dataset was used. Consider distinguishing between $P$ and $Q$ using a prediction rule $\phi : X \to [0, 1]$. The **type I error** (false positive) and **type II error** (false negative) are:

$$\alpha = \mathbb{E}_{x \sim P}[\phi(x)], \qquad \beta = \mathbb{E}_{x \sim Q}[1 - \phi(x)].$$

**Definition 3.2** (Tradeoff Function). The tradeoff function $T(P\|Q) : [0, 1] \to [0, 1]$ captures the optimal type II error achievable for a given type I error bound:

$$T(P\|Q)(\alpha) = \inf_{\phi} \left\{ 1 - \mathbb{E}_Q[\phi] : \mathbb{E}_P[\phi] \leq \alpha \right\}.$$

For the most powerful likelihood-ratio test with threshold $t$, the error rates become:

$$\alpha(t) = \Pr_{X \sim P}[L(X) < t], \qquad 1 - \beta(t) = \Pr_{X \sim Q}[L(X) \geq t].$$

Note that if $T(P, Q) = f$, then $T(Q, P) = f^{-1}$. A tradeoff curve $f$ is called symmetric if $f^{-1} = f$.

**Definition 3.3** ($f$-DP). An algorithm $M$ is $f$-**differentially private** if for every pair of neighboring databases $\mathcal{D}, \mathcal{D}'$:

$$T(M(\mathcal{D})\|M(\mathcal{D}')) \succeq f,$$

where $\succeq$ denotes pointwise ordering.

The classical $(\varepsilon, \delta)$ view is recovered from $L$ via:

$$\delta(\varepsilon) = \Pr[L > \varepsilon] - e^\varepsilon \Pr[L < -\varepsilon].$$

**Privacy Accounting.** A numerical privacy accountant (Gopi et al., 2021) is a procedure that tracks and composes the privacy-loss random variables of randomized mechanisms over multiple executions to compute the overall $(\varepsilon, \delta)$-DP guarantee. It incorporates sub-sampling amplification and composes privacy loss across steps or epochs via convolution. This provides tight privacy guarantees for complex multi-step mechanisms by tracking full distributional information rather than worst-case bounds.

Additional properties such as amplification via sub-sampling and composition for $f$-DP are detailed in Appendix A.1

## 4. Proposed Method

DP-SGD with Fast Gradient Clipping (FGC) or Ghost Clipping (GC) operates as follows: in each iteration, we have,

1. First backward pass computes per-sample gradient norms $\{n_i\}$
2. Loss rescaling (equivalent to gradient clipping): $\widehat{\mathcal{L}}_i = \min \left( \frac{C}{n_i}, 1 \right) \mathcal{L}_i$
3. Second backward pass on the aggregated loss $\widehat{\mathcal{L}} = \sum_i \widehat{\mathcal{L}}_i$ followed by noise addition and optimizer step.

The above implementation improves memory efficiency in many practical settings. This is achieved because (a). it is sufficient to materialize per-sample gradients only one layer at a time, and (b). for *linear-like* layers (eg: linear, attention, convolution), per-sample norm computation is *easier* than per-sample gradients. For an $L$-layer feed-forward network with input/output dimensions $d < p$, context length $T$, and batch size $B$: SOTA (FGC/GC) memory complexity is $O(B \min(d^2, T^2))$. This improves over naive DP-SGD's $O(BLd^2)$ memory, but remains quadratic in input parameters which is prohibitive for large-scale settings.

**Algorithm 1** DP-SGD-RC

**Input:** Dataset $D$, Model parameters $W = \{W_1, W_2, \ldots, W_L\}$ where $L$ is number of layers, Loss function $(W, x) \mapsto \mathcal{L}(w; x)$, Noise multiplier $\sigma$, Clipping threshold $C$, Iterations $T$, Initial $W^0$

**Input:** Norm-Estimation-Routine

1: **for** each $t$ in $T$ **do**
2:     Sample a batch of data $\{x_i\}_{i=1}^B \sim D$
3:     **for** layer $l \in 1, 2, \cdots, L$ **do**
4:         Get activation tensor $\{\boldsymbol{a}_{(l)}^t\}_i$ by Forward hook
5:     **end for**
6:     Compute per-sample loss: $\{\mathcal{L}_i^t\}$
7:     **for** layer $l \in L, L-1, \cdots, 1$ **do**
8:         Get output gradient $\left\{ \frac{\partial \mathcal{L}_i^t}{\partial \boldsymbol{s}_{(l)}^t} \right\}$ by backward hook
9:         Compute per-sample gradient norm $\{(\widehat{n}_i)_l^t\}$ using Norm-Estimation-Routine
10:     **end for**
11:     Sum layer-wise norms: $n_i^t = \left\| \frac{\partial \mathcal{L}_i^t}{\partial \mathbf{W}^t} \right\|_F^2 = \sum_l (\widehat{n}_i)_l^t$
12:     Compute scaled loss: $\widehat{\mathcal{L}}^t = \sum_{i=1}^b \min\left( \frac{C}{\sqrt{n_i^t}}, 1 \right) \mathcal{L}_i^t$
13:     Compute gradient $\nabla^t$ back-propagating through $\widehat{\mathcal{L}}^t$
14:     Add Gaussian noise $\widehat{\nabla}^t = \nabla^t + \sigma C \cdot \mathcal{N}(0, \mathbf{I})$
15:     Apply SGD/Adam with the private gradient $\widehat{\nabla}^t$
16: **end for**

**Output:** $\mathbf{W}^T$

---

**Algorithm 2** Norm-Estimation-Routine

**Input:** $A = \{\boldsymbol{a}_{(l)}^t\}_i \in \mathbb{R}^{B \times T \times d}$, $G = \left\{ \frac{\partial \mathcal{L}_i^t}{\partial \boldsymbol{s}_{(l)}^t} \right\} \in \mathbb{R}^{B \times T \times p}$, $k$, layer-type $\in \{$linear-like, other$\}$, estimator $\in \{$Hutch, Hutch$^{++}\}$; assume $d < p$

1: **for** $i = 1$ to $B$ **do**
2:     **if** layer-type = linear-like **then**
3:         **if** estimator = Hutch **then**
4:             Sample $P \in \mathbb{R}^{p \times k}$ with $P_{uv} \sim \mathcal{N}(0, 1/\sqrt{k})$
5:             $Y_i \leftarrow A_i^\top (G_i P)$
6:             $\widehat{n}_i \leftarrow \|Y_i\|_F^2$
7:         **else if** estimator = Hutch$^{++}$ **then**
8:             Sample $P, S \in \mathbb{R}^{p \times k}$; $P_{uv}, S_{uv} \sim \mathcal{N}(0, 1/\sqrt{k})$
9:             Compute $Q \in \mathbb{R}^{d \times k}$, the orthonormal basis of $\text{Col}(G_i^\top (A_i (A_i^\top (G_i S))))$
10:            $U_i \leftarrow A_i^\top (G_i Q) \in \mathbb{R}^{d \times k}$
11:            $V_i \leftarrow A_i^\top (G_i P) - U_i(Q^\top P) \in \mathbb{R}^{d \times k}$
12:            $\widehat{n}_i \leftarrow \|U_i\|_F^2 + \|V_i\|_F^2$
13:        **end if**
14:    **else**
15:        Compute per-sample gradient, $M_i$, directly
16:        $\widehat{n}_i \leftarrow \|M_i\|_F^2$
17:    **end if**
18: **end for**

**Output:** Per-sample norm-squared estimates for a layer: $\{\widehat{n}_i\}_{i=1}^B$

---

Before outlining our proposed approach, we briefly review the per-sample gradient norm computation used in FGC and GC methods. Let $\mathbf{S} = \mathbf{A}W$ define the linear layer, where $W \in \mathbb{R}^{d \times p}$ is the weight matrix, $\mathbf{A} \in \mathbb{R}^{B \times T \times d}$ is the mini-batch input activations of this layer (a.k.a. the activation tensor), and $\mathbf{S} \in \mathbb{R}^{B \times T \times p}$ is the output. In addition, let $\mathbf{G} = \frac{d\mathcal{L}}{d\mathbf{S}} \in \mathbb{R}^{B \times T \times p}$ denote the loss gradient with respect to the output. The $i$-th example in the per-sample gradient is $A_i^\top G_i$. Since the computation is identical for each instance in the mini-batch, we restrict to activation and gradient matrices, $A$ and $G$ and drop subscripts henceforth. The fast gradient and ghost clipping trick computes per-sample norms as follows,

1. FGC: $n = \|A^\top G\|_F^2$ with space $O(d^2)$
2. GC: $n = \text{vec}(AA^\top)^\top \text{vec}(GG^\top)$ with space $O(T^2)$

We propose DP-SGD-RC leveraging **stochastic trace estimation** for memory-efficient per-sample gradient computation. As a notation shorthand, $((AB)C)$ means compute $AB$ first, then multiply by $C$. Observe that

$$\|A^\top G\|_F^2 = \text{trace}(G^\top A A^\top G) = \text{trace}(O)$$

A line of work on stochastic trace estimation focuses on

estimating trace, without explicit access to $O$. One such approach assumes only matrix-vector product query access to $O$. The query complexity can be roughly translated to corresponding memory complexity and runtime. A simple such estimator is the *Hutchinson's estimator* (Hutchinson, 1989) based on random projections:

$$\text{Hutch}_k(O) = \sum_{i=1}^k P_i^\top O P_i = \text{trace}(P^\top O P) = \left\| (P^\top G^\top)^\top A \right\|_F^2$$

where $P \in \mathbb{R}^{p \times k}$ consists of i.i.d $\mathcal{N}(0, 1/k)$ entries. This admits an implementation with $O(k(p + T + d))$ space. Note that we can equivalently consider $P \in \mathbb{R}^{d \times k}$ and multiply on the "left" but we restrict to $d < p$ assumption for simplicity.

A classical analysis (Hutchinson, 1989; Avron & Toledo, 2011) showed that with $k = O\left( \frac{\log(1/\beta)}{\alpha^2} \right)$, with probability $\geq 1 - \beta$, we have $\text{Hutch}_k(O) \in (1 \pm \alpha)\text{trace}(O) = (1 \pm \alpha) \|A^\top G\|_F^2$. This is akin to the celebrated Johnson-Lindenstrauss (JL) result and its construction (Johnson et al., 1984; Dasgupta & Gupta, 2003).

A recent improvement, Hutch$^{++}$ (Meyer et al., 2021), improves it estimating the *head* of the eigen-spectrum via low-

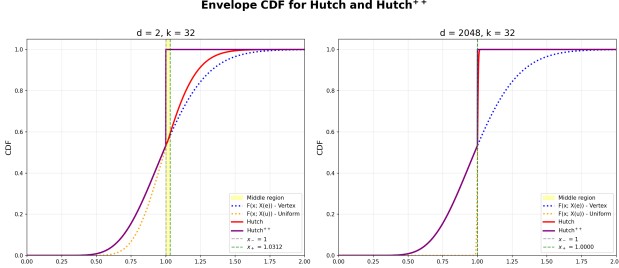

*Figure 1.* Envelope functions for Hutch and Hutch$^{++}$ for $k = 32$ and $d = 2$ (left) and $d = 2048$ (right). In the left, we see a separation between Hutch and Hutch$^{++}$, but in the right (practical setting), they are essentially overlapping.

rank approximation with sketching, while using Hutchinson's estimator for the *tail*.

$$
\begin{aligned}
\mathsf{Hutch}_k^{++}(O) &= \operatorname{trace}(Q^\top O Q) \\
&+ \operatorname{trace}\big(P^\top (I - QQ^\top)^\top O(I - QQ^\top)P\big) \\
&= \left\| (QG^\top)A \right\|_F^2 + \left\| (P((I - QQ^\top)A^\top G)^\top \right\|_F^2 \\
&= \left\| (QG^\top)A \right\|_F^2 + \left\| (PG^\top)A - (((PG^\top)A)Q)Q^\top \right\|_F^2
\end{aligned}
$$

where $P \in \mathbb{R}^{p \times k}$ consists of i.i.d $\mathcal{N}(0, 1/k)$ entries as before, and $Q \in \mathbb{R}^{p \times k}$ is the orthogonal basis for the span of $\operatorname{Col}(OS)$, where $S$ also consists of i.i.d $\mathcal{N}(0, 1/k)$ entries. The implementation requires $O(k^2 + k(T + d + p))$ space. They showed that with $k = O\left(\frac{\sqrt{\log(1/\beta)}}{\alpha}\right)$, with probability $\geq 1 - \beta$, $\mathsf{Hutch}_k^{++}(O) \in (1 \pm \alpha)\operatorname{trace}(O)$, establishing a quadratic improvement on Hutchinson's. Finally, (Meyer et al., 2021) proved a lower bound indicating that $O(1/\alpha)$ matrix-vector products are necessary, nearly achieving the limits of this computational model with $\mathsf{Hutch}^{++}$.

The key idea behind our proposed algorithm, DP-SGD-RC, is simple: replace the exact norm estimation in DP-SGD with a stochastic trace estimation routine. The algorithm 1 is the pseudo-code for DP-SGD with randomized clipping with a provided norm estimation routine. Algorithm 2 outlines the norm estimation routines for Hutch and Hutch$^{++}$.

## 5. Privacy Analysis and Accounting

In this section, we sketch the privacy analysis of the proposed method and efficient accounting. We note that although our theorem statements are expressed in terms of $f$-DP, the final privacy guarantee of interest is the standard $(\epsilon, \delta)$-DP. The $f$-DP formalism serves primarily as an analytical tool for tightly tracking privacy loss under composition, and is used internally by modern privacy accountants for DP-SGD, including the PRV accountant (Gopi et al., 2021),

FFT-based accountants (Koskela et al., 2020), and production libraries such as *Opacus* (Yousefpour et al., 2021; Aketi et al., 2025) and *dp-accounting* (Google Differential Privacy Team, 2020). For our experiments, the resulting $f$-DP guarantees are converted to corresponding $(\epsilon, \delta)$-DP via a numerical conversion pipeline akin to (Gopi et al., 2021), the details are discussed in Section 5.2.

A few remarks on notation: for a random variable $X$, we denote its CDF as $x \mapsto F(x; X)$. Further, we use $\preceq_{st}$, $\preceq_{cvx}$, $\preceq_{sl}$ to denote partial order between random variables in the stochastic dominance, Schur-convexity and stop-loss order sense respectively; these are formally defined in Appendix A.2.

### 5.1. Privacy Analysis

The following is the main result, which characterizes the trade-off curve for a single-step of Algorithm 1 using Algorithm 2 as the norm estimator.

**Theorem 5.1.** *A single step of Algorithm 1 is $f$-DP with* $f = T\left(\frac{1}{Z_{k,d}^2}, \mathcal{N}\left(\frac{1}{\sigma Z_{k,d}^2}, 1\right) \middle\| \frac{1}{Z_{k,d}^2}, \mathcal{N}(0, 1)\right)$ *where $Z_{k,d}$ is identified with its CDF $F(\cdot; Z_{k,d})$ as follows.*

1. **Hutch**: *There exists $x_+ \in [1, 2]$ with $F(x; Z_{k,d}) =$*

$$
\begin{cases}
F\left(x; \frac{1}{kd} \sum_{i=1}^d \chi_k^2\right), & x \leq 1 \\
\sup_{i,j,\lambda: 1 \leq i+j \leq d, \lambda \in [0,1]} F(x; S(i,j,\lambda)) & x \in (1, x_+) \\
F\left(x; \frac{1}{k}\chi_i^2(k)\right) & x \geq x_+
\end{cases}
$$

*where $S(i,j,\lambda) = \frac{\lambda}{ik}\chi^2(ik) + \frac{(1-\lambda)}{jk}\chi^2(jk)$.*

2. **Hutch$^{++}$**: $F(x; Z_{k,d}) = \max(F(x; \frac{1}{k}\chi_1^2(k)), 1_{\geq 1}(x))$

Note that when $Z_{k,d}^2 \equiv 1$, we recover the tradeoff function for DP-SGD. Further, for $d = 1$, Hutch reduces to JL projection yielding $Z_{k,d} \frac{1}{k}\chi^2(k)$ as in DP-SGD-JL (Bu et al., 2021).

**Proof Sketch.** We outline the main steps only for the Hutch case. The full proof is deferred to Appendix B.

It is sufficient to limit to a single *widest* layer. Note that for every linear layer of shape $(p_l, d_l)$, we project along the larger dimension: $P : \mathbb{R}^{p_l \times d_l} \to \mathbb{R}^{k \times d_l}$. Let $d = \max_l \min(p_l, d_l)$ denote that *max-min* layer size. Now, since we sample a fresh random projection for every layer, the operation is equivalent to considering a *stacked* projection matrix $P \in \mathbb{R}^{k \times \sum_l p_l}$ operating on a matrix of shape $\sum_l p_l \times d$. Our analysis (below) is independent of $\sum_l p_l$ yielding the stated sufficiency.

Let $D$ and $D'$ be neighbouring datasets with differing element $(A_0, G_0)$ and let $Q_0 := A_0^\top G_0$. The first step is

Lemma B.2, which shows that for any norm estimation routine, $x \mapsto R(x)$, the trade-off function is bounded as,

$$T(\mathcal{A}(D)\|\mathcal{A}(D')) \succeq T\left(Z, \mathcal{N}(Z,1)\|Z, \mathcal{N}(0,1)\right)$$

where $Z = \frac{\|Q_0\|}{R(Q_0)}$. Recall the Hutch estimator is,

$$\left\|P(A^\top G)^\top\right\|_F^2 = \text{trace}(P^\top O P) = \sum_{i=1}^k P_i^\top O P_i$$

where $O = (A^\top G)(A^\top G)^\top$. For the differing element, $(A_0, G_0)$, the distribution of the summand, $P_i^\top O_0 P_i$ is $\sum_{j=1}^d \lambda_j \chi_j(1)^2$ where $\lambda_j$ are eigenvalues of $O_0$. Further, the distribution of the sum, $\sum_{i=1}^k P_i^\top O P_i$ is the generalized chi-squared distribution $\sum_j \lambda_j \chi_j(k)^2$ with weights $\{\lambda_j\}$. Note,

$$\|Q_0\|_F^2 = \text{trace}((A_0^\top G_0)(A_0^\top G_0)^\top) = \text{trace}(O_0) = \|\lambda\|_1$$

This yields that $Z^2$ (denoted as $Z^2(\lambda)$) is distributed as,

$$Z^2(\lambda) \sim \frac{\|\lambda\|_1}{\sum_i \lambda_i \chi^2(k)}$$

Note that $Z(\lambda)$ is scale-invariant, so it suffices to restrict to the simplex $\lambda \in \Delta^{d-1}$, giving us $Z(\lambda) \sim (\sum_i \lambda_i \chi^2(k))^{-1}$.

The above is still data-dependent (through $\lambda$). For privacy analysis, we need to remove this data-dependence yielding the *dominating pair*. This key step is achieved via Proposition 5.2 (below) giving a data-independent random variable $Y \preceq_{st} \sum_i \lambda_i \chi^2(k) \implies Y^{-1} \succeq_{st} Z(\lambda)$ for all $\lambda \in \Delta^{d-1}$. We finally invoke Lemma B.3 to get

$$
\begin{aligned}
T(\mathcal{A}(D)\|\mathcal{A}(D')) &\succeq T\left(Z, \mathcal{N}(Z,1)\|Z, \mathcal{N}(0,1)\right) \\
&\succeq T\left(Y^{-1}, \mathcal{N}(Y^{-1},1)\|Y^{-1}, \mathcal{N}(0,1)\right)
\end{aligned}
$$

This completes the proof sketch. Below is the key step in the above proof which establishes the *extremal envelope* of convex combination of i.i.d. chi-squared random variables.

**Proposition 5.2.** *Let $X_i \sim \frac{1}{k}\chi^2(k)$ be $d$ i.i.d random variables. For $\lambda \in \Delta^{d-1}$, define $X(\lambda) = \sum_{i=1}^d \lambda_i X_i$. Let $f(x) = \sup_{\lambda \in \Delta^{d-1}} F(x; X(\lambda))$ denote the envelope distribution, where $F(\cdot; X(\lambda))$ denotes the CDF of $X(\lambda)$. There exists $x_+ \in [1, 2]$ such that*

1. *$f(x) = F(x; Z_{k,d})$ as defined in Theorem 5.1*
2. *As $k \to \infty$, $x_+ = 1 + \frac{2}{dk} + O\left(\frac{1}{k^2}\right)$*

We first contrast it with the setup and result in (Bu et al., 2021). In their case, $d = 1$, therefore $\lambda = \lambda_1 = 1$ trivially. Thus, the envelope function is simply $F(x; X_i)$ recovering their result. In our case, the solution is more complicated.

**Proof Sketch.** We outline the major steps in the proof which uses tools from stochastic orders and majorization theory in probability. Additional preliminaries are in Appendix A.2. Let $e = (1, 0, \ldots, 0)$ and $u = (d^{-1}, d^{-1}, \ldots, d^{-1})$ denote the extremal vertex and center configurations respectively.

1. **Stochastic Ordering**: For majorization chain $e \preceq \lambda \preceq u$, we establish *schur-convex* chain $X(e) \preceq_{cvx} X(\lambda) \preceq_{cvx} X(u)$ which futher implies a *stop-loss* chain, $X(e) \preceq_{sl} X(\lambda) \preceq_{sl} X(u)$.

2. **Single-crossing**: For $\lambda \preceq \mu$ with $X(\lambda) \succeq_{sl} X(\mu)$, we have that $X(\lambda)$ and $X(\mu)$ exhibits a *single-crossing* property. This is shown using log-concavity of $X(\lambda)$ which implies *Decreasing Mean Residual Life* property.

3. **Extremal envelopes**: This implies existence of thresholds, $x_-$ and $x_+$ such that below $x_-$, $X(e)$ dominates every $X(\lambda)$, and above $x_+$, $X(u)$ dominates every $X(\lambda)$.

4. **Middle region envelope**: The middle region $(x_-, x_+)$, though narrow, is real and we show in Fig. 8 for $d = 2$ that $\lambda$ switches continuously from 0 to 0.5. Further, (Székely & Bakirov, 2003)[1] showed that

$$\text{Ext}_\lambda F(x; X(\lambda))$$

$$= \underset{i,j,\lambda: 1 \le i+j \le d, \lambda \in [0,1]}{\text{Ext}} F\left(x; \frac{\lambda}{ik}\chi^2(ik) + \frac{(1-\lambda)}{jk}\chi^2(jk)\right)$$

Finally, in Proposition B.9 we show the claimed $x_- = 1$ and establish the asymptotic form of $x_+$.

**Efficient algorithm for envelope.** We focus only on the middle region since extremal parts have explicit form. Note that given a region $\Delta x = (1, x_+)$, we can brute-force the above form with complexity $O(d^2 n_{\Delta x} n_\lambda)$ where $n_\lambda, n_{\Delta x}$ denote associated discretizations. Further, using single crossing between all $X(\lambda)$ and $X(u)$, we can approximate $x_+ \in [1, 2]$ with a simple binary search in $O(d^2 n_\lambda \log(1/\Delta_x))$ time. Finally, all the computations above are massively parallel.

**Analysis of Hutch$^{++}$.** For tractability, we assume that the adversary knows the *head* component of the estimator. This results in a similar problem of stochastically dominating the random variable $\sum_{i=1}^k \lambda_i + k^{-1} \sum_{i=k+1}^d \lambda_i \chi_i^2(k)$. We note that this assumption does not affect our results in practical

---

[1]Theorem 2 in (Székely & Bakirov, 2003) study the exact question as in Proposition 5.2 and establish the three-region envelope. Their claimed middle region is surprisingly simple, with only one non-zero $\lambda$: $\sup_{\lambda \in [0,1]} \mathbb{P}(\lambda X_1 + (1-\lambda)X_2 \le x)$. However, we identify a bug in the proof of the claim (though not in an intermediate result we use) and show that, in fact, the claim is not true from simple simulations (see Appendix B.1 for details).

settings. As demonstrated in Table 4 and Figure 12, the resulting noise multipliers converge to values comparable to the Hutch case when the hidden dimension, $d$, is sufficiently large. Relaxing this assumption and fully characterizing the privacy properties of Hutch$^{++}$ remains an interesting open problem, which we leave for future work. More details are provided in Appendix B.7.2.

### 5.2. Privacy Accounting

We sketch the intuition and mechanics behind a privacy accountant for DP-SGD-RC that incorporates randomized clipping via an *envelope* distribution. Rather than a fixed noise scale in Gaussian mechanism, the DP-SGD-RC averages the single-step Gaussian privacy kernels over a *random effective scale*, $a$, induced by the norm estimation. In particular, $a = \frac{1}{\sqrt{Y}}$ where $Y$ is the envelope random variable. The Gaussian single-step test errors depend on the signal-to-noise ratio (SNR) $a/\sigma$:

$$\alpha(t) = \Phi\Big(-\frac{t}{a/\sigma} - \frac{a/\sigma}{2}\Big), \qquad \beta(t) = \Phi\Big(\frac{t}{a/\sigma} - \frac{a/\sigma}{2}\Big),$$

with $\Phi$ the standard normal CDF. If the envelope puts more mass on *small* $Y$ (thus larger $a$), the effective SNR increases, single-step privacy-loss tails get heavier, and total privacy weakens (larger $\varepsilon$ for the same $\delta$). Conversely, mass on *large* $Y$ (smaller $a$) strengthens privacy.

We compute the expectations in $\alpha$ and $\beta$ by a Riemann–Stieltjes construction when $Y$ is available via its CDF $F(y; Y)$, bypassing the PDF construction which is spiky and numerically problematic in our setting:

$$\alpha(t) = \mathbb{E}_Y\Big[\Phi\Big(-\frac{t}{(1/\sqrt{Y})/\sigma} - \frac{(1/\sqrt{Y})/\sigma}{2}\Big)\Big]$$
$$= \int \Phi\Big(-\frac{t}{a/\sigma} - \frac{a/\sigma}{2}\Big)\, dF(a; A),$$
$$\beta(t) = \mathbb{E}_Y\Big[\Phi\Big(\frac{t}{(1/\sqrt{Y})/\sigma} - \frac{(1/\sqrt{Y})/\sigma}{2}\Big)\Big]$$
$$= \int \Phi\Big(\frac{t}{a/\sigma} - \frac{a/\sigma}{2}\Big)\, dF(a; A),$$

where $F(\cdot; A)$ is the induced CDF of $A = 1/\sqrt{Y}$ under $F(\cdot; Y)$. Numerically, we partition $[y_{\min}, y_{\max}]$, take CDF increments $w_i = F(y_i; Y) - F(y_{i-1}; Y)$, set a representative $\bar{y}_i$ per-bin and $a_i = 1/\sqrt{\bar{y}_i}$, and approximate

$$\alpha(t) \approx \sum_i w_i\, \Phi\Big(-\frac{t}{a_i/\sigma} - \frac{a_i/\sigma}{2}\Big), \beta(t) \approx \sum_i w_i\, \Phi\Big(\frac{t}{a_i/\sigma} - \frac{a_i/\sigma}{2}\Big)$$

We provide the accounting pseudo-code and full details in Appendix B.9. In particular, Appendix B.9.1 details the algorithmic mechanics for computing the envelope CDF

based on our $f$-DP guarantees. We further extend these results in Appendix B.9.2 to incorporate privacy amplification via sub-sampling and composition across time steps, following the numerical approach of Gopi et al. (2021) to build a complete privacy accountant for our mechanism.

## 6. Experiments

In this section, we present a comprehensive evaluation of our proposed methods across multiple tasks and datasets. We focus on three main tasks: classification (BBC (Greene & Cunningham, 2006)), summarization (BillSum (Kornilova & Eidelman, 2019)), and question answering (HotpotQA (Yang et al., 2018)). Our method is most effective when the context length ($T$) exceeds the linear layer dimensions; therefore, we use datasets with context length $\geq 4096$ and fine-tune at a fixed length of $4096$. We fine-tune Llama 3.2 1B using (1) full fine-tuning and (2) LoRA fine-tuning of selected linear layers, and compare DP-SGD-RC with non-private baselines and DP-SGD. For private training, we consider $\varepsilon \in \{0.7, 2, 9\}$, with $\delta = 10^{-5}$ for BBC and $\delta = 10^{-6}$ for BillSum and HotpotQA. Additional details and hyper-parameters are provided in Appendix D.

Table 2 reports results for full fine-tuning of Llama 3.2 1B on the considered tasks. We compute peak memory for each linear layer independently with batch size 2 per GPU. With a projection dimension, $k$, as small as 32, our method achieves performance comparable to DP-SGD for $\varepsilon = 2$ and 9 while reducing peak memory by $15 - 40\%$. On BBC, our method incurs at most a $0.7\%$ accuracy drop relative to DP-SGD; on BillSum and HotpotQA, ROUGE-1 and Exact Match decrease by $0.019$ and $0.04\%$, respectively, indicating minimal utility loss with substantial memory savings. We train each model across 3 independent random seeds and report the mean and standard deviation of the results.

Table 3 presents results on LoRA fine-tuning of Llama 3.2 1B model. We observe $< 0.4\%$, and $0.005$ drop in the corresponding metrics for BBC and BillSum respectively. We note that the LoRA finetuning setup and DP-SGD-RC are orthogonal approaches i.e., LoRA reduces the number of trainable parameters, while DP-SGD-RC reduces the overhead of per-sample gradient norm computation. Since DP-SGD with LoRA is already memory-efficient, DP-SGD-RC's gains are most pronounced in the full fine-tuning setting. The results in Table 3 confirm that DP-SGD-RC remains compatible with LoRA without utility degradation.

Further, we show that using Hutch$^{++}$ to estimate the per-sample norm instead of Hutch can improve the utility in the lower epsilon regime as shown in Table 4. We observe better utility with Hutch$^{++}$ i.e., $\sim 3\%$ improvement in accuracy in BBC dataset. We attribute this to the regularization effect of noisy norm estimates. Note that for $k = 32$, Hutch$^{++}$

*Table 2.* Comparison of the proposed DP-SGD-RC with DP-SGD baseline for full fine-tuning of the Llama 3.2 1B model.

| Dataset | Task (Metric) | Method | Proj Dim. ($k$) | Metrics | |
|---|---|---|---|---|---|
| | | | | $\epsilon = 9$ | $\epsilon = 2$ |
| BBC | Classification (Accuracy ↑) | non-private | N/A | $95.20 \pm 0.51\%$ | |
| | | DP-SGD (FGC) | N/A | $96.33 \pm 0.59\%$ | $94.06 \pm 0.12\%$ |
| | | **DP-SGD-RC (Ours)** | 32 | $\mathbf{96.40 \pm 0.22}\%$ | $\mathbf{95.60 \pm 0.37}\%$ |
| BillSum | Summarization (ROUGE1 ↑) | non-private | N/A | $0.4928 \pm 0.0027$ | |
| | | DP-SGD (FGC) | N/A | $0.4882 \pm 0.0011$ | $0.4831 \pm 0.0005$ |
| | | **DP-SGD-RC (Ours)** | 32 | $0.4864 \pm 0.0013$ | $0.4796 \pm 0.0018$ |
| HotpotQA | Question Answering (Exact-Match ↑) | non-private | N/A | $61.06 \pm 0.39\%$ | |
| | | DP-SGD (FGC) | N/A | $61.44 \pm 0.05\%$ | $61.35 \pm 0.03\%$ |
| | | **DP-SGD-RC (Ours)** | 32 | $61.42 \pm 0.08\%$ | $61.31 \pm 0.09\%$ |

*Table 3.* Comparison of the proposed DP-SGD-RC with DP-SGD baseline for LoRA fine-tuning of Llama 3.2 1B model.

| Dataset | Task (Metric) | Method | Proj Dim. ($k$) | Metrics | |
|---|---|---|---|---|---|
| | | | | $\epsilon : 9$ | $\epsilon : 2$ |
| BBC | Classification (Accuracy ↑) | non-private | N/A | 96.5% | |
| | | DP-SGD (FGC) | N/A | 96.3% | 93.1% |
| | | **DP-SGD-RC (Ours)** | 32 | 95.9% | 93.2% |
| BillSum | Summarization (ROUGE ↑) | non-private | N/A | 0.491 | |
| | | DP-SGD (FGC) | N/A | 0.490 | 0.487 |
| | | **DP-SGD-RC (Ours)** | 32 | 0.488 | 0.486 |

has a similar peak memory reduction but higher latency and compute overhead compared to Hutch. For a fixed epsilon, when the dimension of linear layer is large (i.e., $p, d \gg 128$), the noise multipliers for both Hutch and Hutch$^{++}$ estimation are essentially same with our accountant. This can be seen from Figure 1 where for large $d$ (right), the envelope functions for the two overlap. Our experiments show that Hutch$^{++}$ has an order of magnitude lower error in norm estimation as shown in Figure 11a.

*Table 4.* Hutch and Hutch$^{++}$ comparison on BBC dataset with full fine-tuning. The experiments were run with three random seeds and the mean and standard deviation of the accuracy is reported.

| Method | Proj Dim. ($k$) | Noise Multiplier ($\epsilon : 0.7, \ \delta : 1e^{-5}$) | Accuracy (%) |
|---|---|---|---|
| DP-SGD | N/A | 4.073 | $67.07 \pm 5.26$ |
| DP-SGD-RC w/ Hutch | 32 | 4.354 | $64.29 \pm 6.77$ |
| DP-SGD-RC w/ Hutch$^{++}$ | 32 | 4.354 | $\mathbf{70.59 \pm 3.49}$ |

### 6.1. Memory, Compute and Latency Gains

To understand the memory, compute and latency improvements of the proposed method as compared to DP-SGD baseline, we conduct ablation studies of the following metrics with full fine-tuning experimental setup: 1) percentage reduction in peak memory (Figure 2), 2) percentage reduction in peak memory without inputs (Figure 13), 3) percentage reduction in FLOPs (Figure 3), and 4) percentage reduction in latency or run-time (Figure 4). For all the above metrics, we compare Hutch and Hutch$^{++}$ estimates of per-sample gradient norm to understand the overheads/gains attained from each estimate. In all the plots, we consider three dif-

ferent linear layers for a fixed context length of 4096: a) $2048 \times 2048$, b) $8192 \times 2048$, and c) $2048 \times 512$ which reflect the linear layers from Llama 3.2 1B model architecture. Note that the proposed algorithm projects the larger dimension (i.e, $\max(p, d)$) and so, the overheads/gains attained for a $p \times d$ linear layer is same as the $d \times p$ linear layer. We compute the overheads/gains independently for each variant of linear layers.

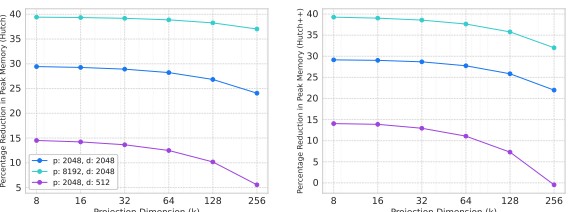

*Figure 2.* Peak memory savings for full fine-tuning settings versus projection dimension for different linear layers of Llama3.2 1B.

Figure 2 shows the reduction in peak memory for a given layer with the proposed method compared to DP-SGD baseline as a function of projection dimension $k$. We observe $39.18\%$ and $38.57\%$ reduction in peak memory for the largest linear layer ($8192 \times 2048$) at $k = 32$ for Hutch and Hutch$^{++}$ respectively. Further, we present reduction in peak memory after removing the memory occupied by inputs (i.e., activations and output gradients) in the appendix (Figure 13). This gives us an estimate of improvements achieved in additional memory required to compute per-sample gradient norm by deploying the proposed method. We observe $99.22\%$ and $97.65\%$ reduction in peak memory without inputs for the largest linear layer ($8192 \times 2048$) at $k = 32$ for Hutch and Hutch$^{++}$ respectively.

For the Hutch estimator, the gains in peak memory vanish as projection dimension $k$ increases to 4096 and 512 for the largest and smallest linear layer respectively. At this juncture, the initialization of the random projection matrix dominates the memory footprint. Note that we sample ele-

ments of the projection matrix from $\mathcal{N}(0, k^{-1})$ instead of $\{-1, 1\}$-valued (Hutchinson, 1989). The latter reduces the memory footprint further as each element requires only 1 bit. We leave this extension for future work.

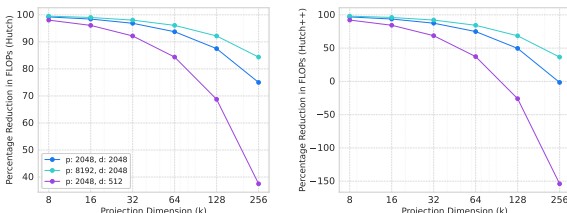

*Figure 3.* Compute savings for full fine-tuning settings versus projection dimension for different linear layers of Llama 3.2 1B Model.

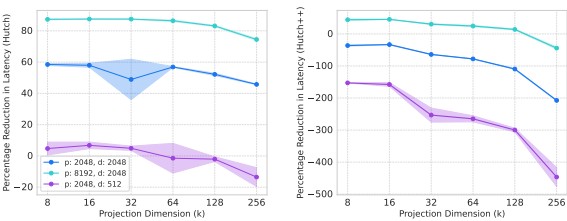

*Figure 4.* Latency savings for full fine-tuning as function of projection dimension for different linear layers of Llama 3.2 1B.

Further, we evaluate the compute and latency overhead/reduction for Hutch and Hutch$^{++}$ estimates compared to DP-SGD baseline. Figure 3 shows percentage reduction in FLOPs (FLoating point OPerations) for Hutch and Hutch$^{++}$ estimates compared to DP-SGD. At a projection dimension of 32, we observe 98.05% and 92.19% reduction in FLOPs with Hutch, and 92.17% and 68.69% reduction in FLOPs with Hutch$^{++}$, for the largest and smallest linear layer respectively. The Hutch estimator reduces the number of computations from $pd$ to $k(d + p)$ $T$-dimensional vector dot products. However, Hutch$^{++}$ has lower compute benefits as it requires 3 times more matrix-vector multiplications than Hutch, including orthogonal basis computation via QR decomposition (Meyer et al., 2021).

Figure 4 presents the percentage reduction latency of the Hutch and Hutch$^{++}$ for a given linear layer on a A100 80GB GPU compared to the baseline DP-SGD. We observe that, for the largest layer, the Hutch estimation is nearly $3\times$ faster than Hutch$^{++}$. This is expected because of the additional sequential operations to accurately estimate the *head* of the eigen spectrum. Overall, for a reasonably small $\epsilon$, say 1.0, DP-SGD-RC with Hutch estimates the norm reasonably well with minimal loss in utility while providing significant reduction in memory footprint, compute and latency.

## 7. Conclusion and Future Work

In this work, we propose DP-SGD-RC, an efficient DP training method for sequential data based on randomized clipping. We provide its privacy analysis and show that it matches DP-SGD in performance while delivering substantial memory and compute savings across several benchmarks. Some immediate future directions include:

1. **Tighter Privacy Analysis for Hutch$^{++}$**: The current privacy analysis of Hutch$^{++}$ assumes that the adversary knows intermediate state of the computation for analytical tractability. This results in *slightly* conservative estimates of noise multipliers for practical settings (large $d$). In the future, we aim to bridge this gap via a tighter analysis.

2. **Sketching Methods**: The use of Gaussian matrices for norm estimation can be replaced with $\{\pm 1\}$-valued matrices, reducing memory and compute costs. Further, we can employ sketching methods that use very sparse matrices and implement matrix-vector multiplication via pseudo-random hashing, yielding additional efficiency gains. This would require extending the privacy analysis to accommodate such techniques.

3. **Improving Memory Complexity**: Explore additional techniques that further improve memory complexity or lower bounds showing otherwise. The trace estimation setting in our context is more structured than standard computational models in the literature. Techniques that sketch both activations and gradients, rather than just one, as is currently done, could be of interest.

4. **Improving Compute Complexity**: Combine DP-SGD-RC with book-keeping techniques (Bu et al., 2023) to further reduce the compute complexity of DP training.

## Impact Statement

This paper presents work whose goal is to advance the field of machine learning. There are many potential societal consequences of our work, none of which we feel must be specifically highlighted here.

## Acknowledgments

We thank Ilya Mironov, Graham Cormode and Will Bullock for constructive suggestions. This work was partially supported by a gift from Meta and a gift from Amazon.

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

# Supplementary Material

## A. Additional Preliminaries

### A.1. Differential Privacy

In this section, we mention some additional properties of tradeoff functions namely, post-processing, composition, and the change in tradeoff function under subsampling. We first start by recalling the definition of tradeoff functions.

Let $P$ and $Q$ be the output distributions of a mechanism on neighboring datasets $\mathcal{D}$ and $\mathcal{D}'$. Consider distinguishing between $P$ and $Q$ using a prediction rule $\phi : X \to [0, 1]$. The **type I error** (false positive) and **type II error** (false negative) are:

$$\alpha = \mathbb{E}_{x \sim P}[\phi(x)], \qquad \beta = \mathbb{E}_{x \sim Q}[1 - \phi(x)].$$

**Definition A.1** (Tradeoff Function). The tradeoff function $T(P\|Q) : [0, 1] \to [0, 1]$ captures the optimal type II error achievable for a given type I error bound:

$$T(P\|Q)(\alpha) = \inf_{\phi} \left\{ 1 - \mathbb{E}_Q[\phi] : \mathbb{E}_P[\phi] \le \alpha \right\}.$$

The tradeoff function provides an operational characterization of hypothesis testing error between two distributions and serves as a fundamental object for analyzing privacy guarantees in the $f$-DP framework. In particular, privacy properties such as post-processing, composition, and subsampling correspond to algebraic operations on tradeoff functions. We summarize these key properties below, which together enable tracking privacy loss under complex randomized mechanisms.

**Proposition A.2** (Post-processing (Dong et al., 2022)). *Let $X, Y$ be two random variables supported on $A$ and let $M : A \to B$ is some randomized function, then $T(X\|Y) \preceq T(M(X)\|M(Y))$.*

**Proposition A.3** (Composition (Dong et al., 2022)). *Let $X_1, X_2, \ldots, X_m$ be independent random variables and let $Y_1, Y_2, \ldots, Y_m$ be independent random variables. Then*

$$T(X_1, X_2, \ldots, X_m \| Y_1, Y_2, \ldots, Y_m) = T(X_1\|Y_1) \otimes T(X_2\|Y_2) \otimes \cdots \otimes T(X_m\|Y_m)$$

*where $\otimes$ is a commutative, associative operation on functions from $[0, 1] \to [0, 1]$.*

For any random variable $X$, we have $T(X\|X) \equiv \mathbb{I}$ where $\mathbb{I} : [0, 1] \to [0, 1]$ is defined as $\mathbb{I}(\alpha) = 1 - \alpha$. The function $\mathbb{I}$ is identity for $\otimes$ operation i.e. $f \otimes \mathbb{I} = f$ for all $f$ (Dong et al., 2022).

**Proposition A.4** (Privacy Curves under Subsampling Dong et al. (2022)). *Let $p \in (0, 1)$ and let $f = T(P\|Q)$. Then $T(P\|(1 - p)P + pQ) = p \cdot f + (1 - p) \cdot \mathbb{I}$.*

We use the following lemma for understanding the behaviour of a trade-off function based on the existence of a coupling which is going to be very useful in our analysis.

**Lemma A.5** (Bu et al. (2021)). *Let $Z, \tilde{Z}$ be two random variables such that there exists some coupling $(Z, \tilde{Z})$ with $Z \ge \tilde{Z} \ge 0$. Then $T(Z, N(Z, 1)\|Z, N(0, 1)) \preceq T(\tilde{Z}, N(\tilde{Z}, 1)\|\tilde{Z}, N(0, 1))$.*

We next recall a useful definition of privacy called Rényi Differential Privacy (RDP). It provides a simple, composable, and analytically tractable way to quantify privacy loss, making it well-suited for iterative mechanisms.

**Definition A.6** (Rényi Differential Privacy (Mironov, 2017)). Let $\alpha > 1$. A randomized mechanism $\mathcal{M}$ satisfies $(\alpha, \varepsilon)$-Rényi Differential Privacy if for all neighboring datasets $D$ and $D'$,

$$D_\alpha(\mathcal{M}(D) \| \mathcal{M}(D')) \le \varepsilon,$$

where the Rényi divergence of order $\alpha$ between two distributions $P$ and $Q$ is defined as

$$D_\alpha(P\|Q) = \frac{1}{\alpha - 1} \log \left( \mathbb{E}_{x \sim Q} \left[ \left( \frac{P(x)}{Q(x)} \right)^\alpha \right] \right).$$

## A.2. Stochastic Orders and Majorization

**Notation.** We use $\prec$ and $\succ$ between vectors to define majorization ordering. We use $\prec_{st}, \prec_{cvx}, \prec_{sl}$ (and $\succ_{st}, \succ_{cvx}, \succ_{sl}$) between probability distributions (with abuse of notation, random variables) to denote partial order based on stochastic dominance, Schur-convexity and stop-loss (defined later in the section)

**Definition (Majorization).** For $x, y \in \mathbb{R}^d$, write $x^{\downarrow}$ for the nonincreasing rearrangement of the coordinates of $x$. We say that $x$ *majorizes* $y$, written $x \succeq y$, if

$$\sum_{i=1}^{k} x_i^{\downarrow} \geq \sum_{i=1}^{k} y_i^{\downarrow} \quad \text{for } k = 1, \ldots, d-1, \quad \text{and} \quad \sum_{i=1}^{d} x_i^{\downarrow} = \sum_{i=1}^{d} y_i^{\downarrow}.$$

Equivalently, from Hardy–Littlewood–Pólya (Marshall et al. (1979), Theorem 2.B.2), $x \succeq y$ iff $y = Tx$ for some doubly stochastic matrix $T$ which is a convex combination of permutation matrices (Birkhoff, 1946).

**Definition (Schur-convexity).** A function $f : \mathbb{R}^d \to \mathbb{R}$ is *Schur-convex* if $x \succeq y$ implies $f(x) \geq f(y)$.

**Lemma A.7** (Symmetric convex $\Rightarrow$ Schur-convex). *If $G : \mathbb{R}^d \to \mathbb{R}$ is convex and permutation-invariant (symmetric), then $G$ is Schur-convex: $x \succeq y \Rightarrow G(x) \geq G(y)$.*

*Proof.* Let $x \succeq y$. By Hardy–Littlewood–Pólya (Marshall et al., 1979) and Birkhoff's theorem (Birkhoff, 1946), there exist permutation matrices $P_1, \ldots, P_m$ and weights $w_1, \ldots, w_m \geq 0$ with $\sum_{j=1}^{m} w_j = 1$ such that $y = \sum_{j=1}^{m} w_j P_j x$. By convexity of $G$,

$$G(y) = G\left(\sum_{j=1}^{m} w_j P_j x\right) \leq \sum_{j=1}^{m} w_j G(P_j x).$$

By symmetry of $G$, $G(P_j x) = G(x)$ for all $j$, hence

$$G(y) \leq \sum_{j=1}^{m} w_j G(x) = G(x).$$

Therefore $G(x) \geq G(y)$ whenever $x \succeq y$, i.e., $G$ is Schur-convex. $\qquad\square$

**Definition (Log-concave density).** A density $f : \mathbb{R} \to [0, \infty)$ is *log-concave* if $\log(f)$ is concave, equivalently

$$f(\theta x + (1-\theta)y) \geq f(x)^{\theta} f(y)^{1-\theta} \quad \text{for all } x, y \in \mathbb{R}, \ \theta \in [0, 1].$$

**Definition A.8** (Survival function). For a distribution function $F$ with density $f$, the survival function is $S(t) = 1 - F(t)$.

**Definition A.9** (Hazard Rate). For a distribution function $F$ with density $f$, the *hazard rate* is $h(t) = \dfrac{f(t)}{S(t)}$, defined for $t$ where $S(t) > 0$.

**Definition A.10** (Mean-Residual Life (MRL)). For a distribution function $F$ with Survial function $S$, the *mean residual life (MRL)* at age $t$ is

$$m(t) = \mathbb{E}[X - t \mid X > t] = \frac{\int_{t}^{\infty} S(u)\, du}{S(t)}.$$

**Definition A.11** (Decreasing Mean Residual Life (DMRL)). For a distribution function $F$, its law is *DMRL (decreasing mean residual life)* if its mean residual life, $m$, is non-increasing in its support.

**Definition A.12** (Increasing Failure Rate (IFR)). For a distribution with hazard rate $h$, it law is *IFR (increasing failure rate)* if $h$ is non-decreasing on its support.

**Definition A.13** (Convex Order). For integrable random variables $X, Y$, we write $X \preceq_{\mathrm{cvx}} Y$ if

$$\mathbb{E}[\varphi(X)] \leq \mathbb{E}[\varphi(Y)] \quad \text{for all convex functions } \varphi \text{ for which the expectations exist.}$$

Note that this implies that $\mathbb{E}[X] = \mathbb{E}[Y]$ by considering $\psi(x) = x$ for one direction and $\psi(x) = -x$ for the other.

**Definition A.14** (Stop–loss order). For integrable random variables $X, Y$, we say $X \preceq_{sl} Y$ if

$$\mathbb{E}\big[(X-t)_+\big] \ \leq \ \mathbb{E}\big[(Y-t)_+\big] \quad \text{for all } t \in \mathbb{R},$$

where $(x)_+ = \max\{x, 0\}$.

The observation $\mathbb{E}\big[(X-t)_+\big] \ = \ \int_t^\infty \big(1 - F(u; X)\big)\, du$ yields the following result.

**Lemma A.15** (Stop loss $\iff$ Integrated CDF order (Layer cake identity)). *For integrable random variables $X, Y$, with CDFs, $F(\cdot; X), F(\cdot; Y)$ and survival functions $S(\cdot, X)$ and $S(\cdot, Y)$ respectively. $X \leq_{sl} Y$ iff*

$$\int_t^\infty S(u; X)\, du \ \leq \ \int_t^\infty S(u; Y)\, du$$

$$\iff \int_{-\infty}^t F(u; X)\, du \ \leq \ \int_{-\infty}^t F(u; Y)\, du, \quad \text{for all } t \in \mathbb{R}.$$

**Lemma A.16** (Convex order $\Rightarrow$ stop loss / integrated–CDF order). *Let $X, Y$ be integrable random variables with distribution functions $F(\cdot; X), F(\cdot; Y)$. If $X \preceq_{cvx} Y$, then $X \preceq_{sl} Y$.*

*Proof.* The proof follows from the fact that $\varphi(x) = (x - t)_+$ is a convex function for any fixed $t \in \mathbb{R}$, thus convex-order implies stop-loss order. $\qquad\square$

**Lemma A.17** (Schur-Ostrowski Criterion). *Let $g : \mathbb{R}^d \to \mathbb{R}$ be continuously differentiable and symmetric (i.e., invariant under permutations of coordinates). Then $g$ is Schur-convex if and only if for all $i \neq j$:*

$$(\lambda_i - \lambda_j)\left(\frac{\partial g}{\partial \lambda_i} - \frac{\partial g}{\partial \lambda_j}\right) \geq 0.$$

**Lemma A.18.** *(Shaked & Shanthikumar, 2007) For a distribution with log-concave density $f$, we have*

1. *$f$ is unimodal and continuous,*

2. *The survival $S$ is log–concave*

3. *It satisfies IFR*

4. *It satisfies DMRL*

## B. Theory

### B.1. Discussion of the result of (Székely & Bakirov, 2003)

The paper of (Székely & Bakirov, 2003) studies extremal probabilities, infimum and supremum, of Gaussian quadratic forms: Theorem 1 and 2 in their paper respectively. We find that the proof and result of Theorem 2 is incorrect. We discuss the mistake and give a counter-example to show the stated result doesn't hold.

The error is located in Section 3, Proof of Theorem 2 (Page 193), in the very first sentence: "By the proof of Theorem 1 ... so all we need to check is the equality"

$$S(x) = \sup_{\lambda \in \Delta^{d-1}} \mathbb{P}\left(\sum_{i=1}^d \lambda_i \chi_i^2(1) \leq x\right) = \max\left(s(x), \max_{1 \leq i \leq d} \mathbb{P}\left(\frac{1}{d}\chi^2(d) \leq x\right)\right),$$

where $s(x) = \sup_{\lambda \in [0,1]} \mathbb{P}(\lambda X_1 + (1 - \lambda)X_2 \leq x)$. The authors invalidly assume that the set of candidate quadratic forms for the Supremum is the same as the set derived for the Infimum in Theorem 1. In Theorem 1 (Infimum): The authors used a perturbation argument (Page 193, Equation 4) to show that if a cluster of eigenvalues has multiplicity $k > 1$, one can "split" them to decrease the probability. This correctly forced the Infimum to the boundary where $k = 1$ (i.e., the rank-2 forms, $s(x)$). In Theorem 2 (Supremum): The authors apply this same restriction to the Supremum. This is not rigorously established. By discarding forms with clustered eigenvalues (e.g., $k = 1, m = n - 1$), the proof ignores the configurations that actually maximize the probability.

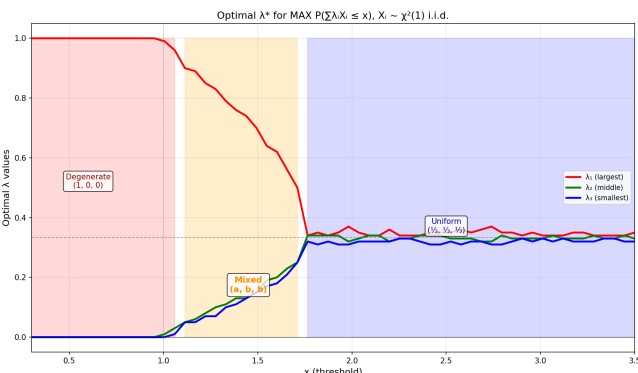

*Figure 5.* Behavior of optimal $\{\lambda_i\}_i$'s for $d = 3$ as a function of $x$

**Numerical Counter-Example.** We can disprove the theorem with the case $d = 3$ and $x = 1.5$. According to Theorem 2, the maximum probability $S_3(1.5)$ is attained either by the equal-weights form or the rank-2 form $s(x)$. Equal Weights gives us:

$$P\left(\frac{1}{3}\chi_3^2 \leq 1.5\right) = P(\chi_3^2 \leq 4.5) \approx \mathbf{0.7877}$$

Rank-2 Max gives us:

$$s(1.5) = \sup_\lambda P(\lambda Z_1^2 + (1 - \lambda)Z_2^2 \leq 1.5) \approx \mathbf{0.7872}$$

The theorem claims the global maximum is $\approx 0.7877$. The True Supremum: Consider a "mixed" quadratic form with multiplicities $k = 1, m = 2$. Let $Q_{mix} = 0.71Z_1^2 + 0.145(Z_2^2 + Z_3^2)$ (note: weights sum to 1).

$$S(1.5) \approx \mathbf{0.7961}$$

We demonstrate this for all values of $x$ in Figure 5. Note that as $\lambda_1$ decreases, both $\lambda_2 \approx \lambda_3$ increase till they yield the uniform distribution. We note that while the existence of more that two non-zero $\lambda_i$ invalidates the claim of (Székely & Bakirov, 2003), it is an evidence for their immediate result that the number of unique $\lambda_i$ is at most 2.

## B.2. Impossibility of RDP Guarantees for Randomized Clipping

In this section, we show that for the case of $d = 1$, we cannot get any RDP guarantees for randomized clipping with Gaussian random matrices. Before that, we would like to state a well-known relation between the Privacy Loss random variable and Rényi Divergence.

For two neighboring datasets $X, X'$, define the privacy loss random variable

$$L := \log\left(\frac{\partial M(X)}{\partial M(X')}\right),$$

where the randomness is over the output of the mechanism $M(X)$.

**Rényi divergence in terms of privacy loss.** The order-$\alpha$ Rényi divergence between $M(x)$ and $M(x')$ is defined as

$$D_\alpha(M(X) \| M(X')) = \frac{1}{\alpha - 1} \log\left(\mathbb{E}_{M(X')}\left[\left(\frac{\partial M(X)}{\partial M(X')}\right)^\alpha\right]\right).$$

Substituting the definition of the privacy loss random variable yields

$$\left(\frac{\partial M(X)}{\partial M(X')}\right)^\alpha = e^{\alpha L}.$$

Moreover, changing the measure from $M(X')$ to $M(X)$ gives

$$\mathbb{E}_{M(X')}\left[e^{\alpha L}\right] = \mathbb{E}_{M(X)}\left[e^{(\alpha - 1)L}\right].$$

Therefore,

$$D_\alpha(M(x) \| M(x')) = \frac{1}{\alpha - 1} \log \left( \mathbb{E}_{M(x)} \left[ e^{(\alpha-1)L} \right] \right).$$

Consider the mechanism which assumes the presence of clipping for every gradient $M(g_1, \cdots, g_n) = \sum_{j=1}^n \frac{\sqrt{k}}{\|g_j\| Z_j} g_j + N$ where $Z_1, \cdots, Z_c \sim \chi(k)$ and $N \sim \mathcal{N}(0, \sigma^2)$.

**Proposition B.1.** *There exist adjacent datasets* $g_1, \cdots, g_c$ *and* $g_1, \cdots, g_c, g_{c+1}$ *for some constant c such that* $D_\alpha(M(g_1, \cdots, g_c, g_{c+1}) \| M(g_1, \cdots, g_c)) = \infty$ *for all* $\alpha \in (1, \infty)$.

*Proof.* Take $M(X) = f(X) + N$ where $N \sim \mathcal{N}(0, \sigma^2)$ and $f$ is potentially randomized. To compute the Rényi divergence, one needs to compute $\mathbb{E}\left[ e^{(\alpha-1)L} \right]$ where $L$ is the privacy loss random variable. Conditioning on $f$, $L$ can be written as $\mathcal{N}(a, b)$ (Kamath, 2020) where

$$a = \frac{\|f(X) - f(X')\|^2}{2\sigma^2} \quad \text{and} \quad b = \frac{\|f(X) - f(X')\|^2}{\sigma^2}.$$

It implies that

$$\begin{aligned} \mathbb{E}\left[ e^{(\alpha-1)L} \right] &= \mathbb{E}\left[ \mathbb{E}\left[ e^{(\alpha-1)L} \mid \|f(X) - f(X')\|^2 \right] \right] \\ &= \mathbb{E}\left[ \exp \left( \frac{\alpha(\alpha-1)}{2\sigma^2} \|f(X) - f(X')\|^2 \right) \right]. \end{aligned}$$

If we consider Hutchinson's via Gaussian projections (which is what we have been considering across the paper), we have the following sensitivity bound

$$\|f(X) - f(X')\|^2 \leq \frac{k}{Z},$$

where $Z \sim \chi^2(k)$.

One can achieve the sensitivity upper bound by considering $X = \{g_1, \cdots g_c\}$ and $X' = \{g_1, \cdots g_c, g_{c+1}\}$ such that $\|g_{c+1}\| = 1$.

Hence, we can get the Rényi divergence by computing the following quantity:

$$\mathbb{E}\left[ e^{(\alpha-1)L} \right] = \mathbb{E}\left[ \exp \left( \frac{\alpha(\alpha-1)k}{2\sigma^2} \cdot \frac{1}{Z} \right) \right], \quad \text{where } Z \sim \chi^2(k),$$

Putting $t = \frac{k\alpha(\alpha-1)}{2\sigma^2}$ and using the expression of the inverse-chi squared distribution (Bernardo et al., 1994)

$$p(x; k) = \frac{2^{-k/2}}{\Gamma(k/2)} x^{-k/2-1} e^{-1/(2x)},$$

we can write out the expectation to be

$$\mathbb{E}\left[ e^{tY} \right] = \frac{2^{-k/2}}{\Gamma(k/2)} \int_0^\infty e^{tx - \frac{1}{2x}} x^{-k/2-1} \, dx.$$

Over the domain $[0, \infty)$, we get the derivative of the integrand $g(x) = e^{tx - \frac{1}{2x}} x^{-k/2-1}$ would be

$$\frac{\partial g}{\partial x} = \left( tx + \frac{1}{2x} - \left( \frac{k}{2} + 1 \right) \right) x^{\frac{-k}{2}-2} e^{tx - \frac{1}{2x}}. \tag{2}$$

Note that, over the domain $[0, \infty)$, we have that $g(x) \geq 0$. Since we have assumed that $\alpha > 1$, we get that $t > 0$. Take $R = \frac{(k+2)+\sqrt{(k+2)^2-8t}}{4t} < \infty$. For each $k$, we get that given $x \geq 0$ $g(x) \geq 0$ and for $x > R$, $\frac{\partial g}{\partial x} > 0$. This implies that $g(x) > g(R)$ for all $x > R$. Thus, we can simply lower bound the integral by $\int_0^\infty g(x) dx \geq \int_R^\infty g(x) dx > \int_R^\infty g(R) dx$ which is infinity. Hence, the above integral evaluates to $\infty$. □

Through the above proposition we get that for $\alpha > 1$, there always exist datasets for which we cannot get any reasonable RDP guarantee removing any possibility of CDP (Bun & Steinke, 2016) and t-CDP (Bun et al., 2018) guarantees as well.

## B.3. Useful results and their extensions from prior work

**Lemma B.2.** *For algorithm with norm estimation routine $x \mapsto R(x)$, given two neighboring datasets, $D$ and $D'$. with differing point $g_0$, we have*

$$T(\mathcal{A}(D)\|\mathcal{A}(D')) \succeq T\left(Z, \mathcal{N}(Z, 1)\|Z, \mathcal{N}(0, 1)\right)$$

*where $Z = \frac{\|g_0\|}{R(g_0)}$.*

*Proof.* The proof follows from Lemma 4.3 of (Bu et al., 2021) observing that the steps up to the above conclusion do not use the form of the norm estimation routine. $\square$

**Lemma B.3.** *For non-negative random variables, $X, Y$*

$$X \prec_{st} \text{Y} \implies T(Y, \mathcal{N}(Y, 1)\|Y, N(0, 1)) \prec T(X, \mathcal{N}(X, 1)\|X, \mathcal{N}(0, 1))$$

*Proof.* The proof uses Lemma A.5, weakening its premising from existence of a coupling $(X, Y)$ with $X \geq Y \geq 0$ to stochastic dominance. In particular, for $X \prec_{st} \text{Y}$, we construct the coupling in the following way

1. Sample $u \sim \text{Unif}(0, 1)$

2. $x = F^{-1}(u; X)$ and $y = F^{-1}(u; Y)$

$(x, y)$ is a valid coupling from inverse transform property of CDF (Strassen, 1965), and $x \geq y$ from $X \prec_{st} Y$, the cdf $F(\cdot; X)$ uniformly dominates $F(\cdot; Y)$. Moreover, the non-negativity of $Y$, implies $X \geq Y \geq 0$. Thus, invoking Lemma A.5 proves the claim. $\square$

## B.4. Key Lemmas for proof of Proposition 5.2

**Lemma B.4** (Single crossing from stop–loss, equal means, and log–concavity). *Let $X, Y$ be nonnegative, absolutely continuous random variables with log–concave densities $f_X, f_Y$ and equal means $\mathbb{E}[X] = \mathbb{E}[Y]$. If $X \succeq_{\text{sl}} Y$, then the CDFs have the single–crossing property: there exists $t^\star \in [0, \infty)$ such that*

$$F_X(t) \geq F_Y(t) \quad \text{for } t \leq t^\star, \qquad F_X(t) \leq F_Y(t) \quad \text{for } t \geq t^\star,$$

*and, unless $F_X \equiv F_Y$, the crossing at $t^\star$ is unique.*

*Proof.* Define the CDF difference and its integral

$$D(t) := F(t; Y) - F(t; X), \qquad G(t) := \int_0^t D(u)\, du.$$

By the equivalence "stop–loss + equal means $\Leftrightarrow$ integrated–CDF order", we have

$$G(t) \leq 0 \quad \text{for all } t \geq 0, \qquad \int_0^\infty D(u)\, du = \mathbb{E}[Y] - \mathbb{E}[X] = 0.$$

Since $G$ is absolutely continuous with derivative $G'(t) = D(t)$ a.e., $G(t) \leq 0$ implies there exists $\varepsilon > 0$ such that $D(t) \leq 0$ for $t \in [0, \varepsilon]$ (otherwise $G$ would locally increase above 0). Because $\int_0^\infty D = 0$ and $D \not\equiv 0$ unless $F(\cdot; X) \equiv F(\cdot; Y)$, there must exist $T > \varepsilon$ such that $D(t) \geq 0$ on a set of positive measure in $(T, \infty)$. Hence at least one sign change occurs.

We now prove *uniqueness* of the sign change. Introduce the tail–gap function

$$H(t) := \int_t^\infty \left(\text{S}(u; Y) - \text{S}(u; Y)\right) du = \mathbb{E}\left[(Y - t)_+\right] - \mathbb{E}\left[(X - t)_+\right] \geq 0 \quad \text{for all } t \geq 0,$$

whose derivative satisfies

$$H'(t) \ = \ -\big(S(t;Y) - S(t;Y)\big) \ = \ -\big(1 - F(t;Y) - (1 - F(t;X))\big) \ = \ -D(t) \quad \text{a.e.}$$

Let $a$ be the (first) crossing point of the survivals (equivalently CDFs), so $S(a;X) = S(a;Y)$ and $D(a) = 0$. Then

$$H(a) \ = \ S(a;Y)\, m_Y(a) - S_X(a)\, m_X(a) \ = \ S(a;X)\big(m_Y(a) - m_X(a)\big) \ \geq \ 0,$$

which implies $m_Y(a) \geq m_X(a)$. Since log–concavity yields DMRL ($m_X, m_Y$ nonincreasing), we have

$$m_Y(t) - m_X(t) \ \geq \ 0 \quad \text{for all } t \geq a.$$

Suppose, for contradiction, that $D$ changes sign at least twice: pick $a < b$ with

$$D(t) \leq 0 \text{ on } [0, a], \quad D(t) \geq 0 \text{ on } [a, b], \quad D(t) \leq 0 \text{ on } [b, c] \text{ for some } c > b.$$

On $(a, b)$, $D \geq 0$ gives $H'(t) = -D(t) \leq 0$, so $H$ is nonincreasing; on $(b, c)$, $D \leq 0$ gives $H'(t) \geq 0$, so $H$ is nondecreasing. Using the tail integral representation $H(t) = S(t;Y)m_Y(t) - S(t;X)m_X(t)$ and the facts

$$S(t;Y) \leq S(t;Y) \text{ on } (a, b), \qquad m_Y(t) \geq m_X(t) \text{ on } [a, \infty),$$

we obtain $H(t) \leq 0$ on $(a, b)$, which contradicts $H(t) \geq 0$ for all $t$. Therefore $D$ cannot exhibit a "$- + -$" pattern. Since $D$ starts non-positive and must eventually be nonnegative, it can change sign at most once, from negative to positive. This yields a unique crossing point $t^\star$ and the stated single–crossing inequalities. $\qquad\square$

**Lemma B.5.** *Let $X_i \sim \frac{1}{k}\chi^2(k)$ be $d$ i.i.d. random variables with $k \geq 2$. For $\lambda, \mu \in \Delta^{d-1}$ with $\lambda \succeq \mu$, define $X(\lambda) = \sum_{i=1}^d \lambda_i X_i$. Then*

$$X(\mu) \preceq_{cvx} X(\lambda).$$

*In particular, with $e = (1, 0, \dots, 0)$ be a vertex of the simplex and $u = \left(\frac{1}{d}, \dots, \frac{1}{d}\right)$ the balanced vector. Then*

$$X(u) \ \preceq_{cvx} \ X(\lambda) \ \preceq_{cvx} \ X(e).$$

*Proof of Lemma B.5.* Follows from Lemma B.6 below. $\qquad\square$

**Lemma B.6.** *[Schur-convexity of weighted sum] Let $Z_1, \dots, Z_d$ be i.i.d. real-valued random variables and let $\varphi : \mathbb{R} \to \mathbb{R}$ be a convex function such that $\mathbb{E}\big[|\varphi(\sum_{i=1}^d a_i Z_i)|\big] < \infty$ for all $a \in \mathbb{R}^d$ in the domain of interest. Define*

$$Q(a) \ := \ \mathbb{E}\left[\varphi\left(\sum_{i=1}^d a_i Z_i\right)\right], \qquad a = (a_1, \dots, a_d) \in \mathbb{R}^d.$$

*The function $Q : \mathbb{R}^d \to \mathbb{R}$ is Schur-convex.*

*Proof.* We first show that $Q$ is symmetric and convex in $a$, and then invoke the fact that any symmetric convex function is Schur-convex.

*Symmetry.* Let $P$ be any permutation matrix on $\mathbb{R}^d$. Since the $Z_i$ are i.i.d., the random variables $\sum_{i=1}^d a_i Z_i$ and $\sum_{i=1}^d (Pa)_i Z_i$ have the same distribution (rename indices). Hence $Q(Pa) = Q(a)$ for all permutations $P$, i.e., $Q$ is permutation invariant (symmetric).

*Convexity.* Fix $a, b \in \mathbb{R}^d$ and $\lambda \in [0, 1]$. By linearity,

$$\sum_{i=1}^d \big(\lambda a_i + (1 - \lambda)b_i\big)Z_i \ = \ \lambda \sum_{i=1}^d a_i Z_i + (1 - \lambda)\sum_{i=1}^d b_i Z_i.$$

By convexity of $\varphi$,

$$\varphi\left(\lambda \sum_i a_i Z_i + (1 - \lambda)\sum_i b_i Z_i\right) \ \leq \ \lambda\, \varphi\left(\sum_i a_i Z_i\right) + (1 - \lambda)\,\varphi\left(\sum_i b_i Z_i\right),$$

and taking expectations yields

$$Q\big(\lambda a + (1-\lambda)b\big) \ \leq \ \lambda Q(a) + (1-\lambda)Q(b).$$

Thus $Q$ is convex. Since $Q$ is symmetric and convex. By Lemma A.7, $Q$ is Schur-convex. $\qquad\square$

**Theorem B.7** (Single crossing for the weighted $\chi^2$ family). *Fix $k \geq 2$ and $d \geq 1$. If $\lambda, \mu \in \Delta^{d-1}$ with $\lambda \succeq \mu$, then the CDFs of $X(\lambda)$ and $X(\mu)$ cross exactly once: there exists a unique $t^\star \in (0, \infty)$ such that*

$$F\big(t; X(\mu)\big) \ \geq \ F\big(t; X(\lambda)\big) \quad \text{for } t \leq t^\star, \qquad F\big(t; X(\mu)\big) \ \leq \ F\big(t; X(\lambda)\big) \quad \text{for } t \geq t^\star.$$

*Proof. Equal means and log–concavity.* Since $\sum_i \lambda_i = \sum_i \mu_i = 1$, we have $\mathbb{E}[X(\lambda)] = \mathbb{E}[X(\mu)] = 1$. For $k \geq 2$, each $X_i$ has a log–concave density; positive scaling preserves log–concavity, and convolution preserves log–concavity. Hence $X(\lambda)$ and $X(\mu)$ have log–concave densities.

*Stop–loss chain via Schur–convexity.* From Lemma B.5, we have that, $X(\mu) \leq_{\mathrm{cvx}} X(\lambda)$, which with Lemma A.16, $X(\mu) \preceq_{\mathrm{sl}} X(\lambda)$.

*Single crossing.* By the lemma B.4, "stop–loss + equal means + log–concavity $\Rightarrow$ single crossing", the CDF difference

$$D(t) \ := \ F\big(t; X(\lambda)\big) - F\big(t; X(\mu)\big)$$

has exactly one sign change: it is $\leq 0$ for small $t$, becomes $\geq 0$ beyond a unique $t^\star$, and never changes back. This yields the stated single–crossing property. $\qquad\square$

**Corollary B.8** (Extremal comparisons on the simplex). *For $k \geq 2$ and $\lambda \in \Delta^{d-1}$, each adjacent pair $(X(u), X(\lambda))$ and $(X(\lambda), X(e))$ has a unique CDF crossing as in Theorem B.7.*

## B.5. Proof of Proposition 5.2

From Theorem B.7 (and Corollary B.8), we have that for every $\lambda, \mu \in \Delta^{d-1}$, the pair $(X(\lambda), X(\mu))$ admit a unique CDF crossing, i.e. $x_{\lambda,\mu} : F(\cdot; X(\lambda)) - F(\cdot; X(\mu))$ crosses at $x_{\lambda,\mu}$.

Let $e = (1, 0, \ldots, 0)$ be a vertex of the simplex and $u = \left(\frac{1}{d}, \ldots, \frac{1}{d}\right)$ the balanced vector, define

$$x^\star := x_{e,u}, \qquad x_- := \min_{\lambda \in \Delta^{d-1}} x_{e,\lambda}, \qquad x_+ := \max_{\lambda \in \Delta^{d-1}} x_{\lambda,u}.$$

Further, we argue that $x_- \leq x^\star \leq x_+$. Taking $\lambda = u$ gives $X(\lambda) = U$, hence $x_{e,u} = x^\star$. Since $x_- = \inf_\lambda x_{e,\lambda} \leq x_{E,u}$, we get $x_- \leq x^\star$. Taking $\lambda = e$ gives $X(\lambda) = E$, hence $x_{e,u} = x^\star$. Since $x_+ = \sup_\lambda x_{\lambda,u} \geq x_{e,U}$, we get $x^\star \leq x_+$.

This gives us that there exists $x_- \leq x^* \leq x_+$ such that

$$F\big(x; X(u)\big) \ \leq \ F\big(x; X(\lambda)\big) \ \leq \ F\big(x; X(e)\big) \quad \text{for } x \leq x_-,$$
$$F\big(x; X(u)\big) \ \geq \ F\big(x; X(\lambda)\big) \ \geq \ F\big(x; X(e)\big) \quad \text{for } x \geq x_+,$$

This gives us the envelope function for the extremal, left and right regions. The middle region is however tricky – the optimal $\lambda$ is not extremal but moves continuously from vertex to the balanced vector (see Fig. Figure 8). To analyze the middle region, we use the structural result in the proof of Theorem 1 in (Székely & Bakirov, 2003), which gives us,

$$\mathrm{Ext}_\lambda F(x; X(\lambda)) = \mathop{\mathrm{Ext}}_{i,j,\lambda:1 \leq i+j \leq d, \lambda \in [0,1]} F\left(x; \frac{\lambda}{ik}\chi^2(ik) + \frac{(1-\lambda)}{jk}\chi^2(jk)\right)$$

Finally, in Proposition B.9, we establish that $x_= 1$ and $x_+ \in [1, 2]$ which completes the proof.

### B.6. Analysis of the middle region of the envelope in Proposition 5.2

In this section, we analyze the middle region of the envelope. We give (non-asymptotic and asymptotic) estimates on the thresholds $x_-$ and $x_+$. We also run simulations for various settings and report values of $x_-$ and $x_+$.

**Proposition B.9** (Extremal crossing points, $x_-$ and $x_+$)**.** *Let $X_1, \ldots, X_d$ be i.i.d. with $X_i \sim \frac{1}{k}\chi^2(k)$. For $\lambda \in \Delta^{d-1} :=$ $\{\lambda \in [0,1]^d : \sum_{i=1}^d \lambda_i = 1\}$ define $X(\lambda) := \sum_{i=1}^d \lambda_i X_i$, and assume that for every $\lambda, \mu \in \Delta^{d-1}$ the difference $x \mapsto F(x; X(\lambda)) - F(x; X(\mu))$ exhibits a unique crossing at $x_{\lambda,\mu} \in (0, \infty)$. Let $e = (1, 0, \ldots, 0)$ and $u = (1/d, \ldots, 1/d)$, and define*

$$x_- := \inf_{\lambda \in \Delta^{d-1}} x_{e,\lambda}, \qquad x_+ := \sup_{\lambda \in \Delta^{d-1}} x_{\lambda,u}.$$

*Then:*

1. *$x_- = 1$ for all $k$ and $d > 1$.*

2. *$x_+ \in [1, 2]$ for all $k$ and $d > 1$. Further, $x_+ = 1 + \frac{2}{dk} + O\left(\frac{1}{k^{-2}}\right)$ for all $d > 1$ as $k \to \infty$*

*Proof.* The two items follow from Proposition B.10 and Proposition B.22 respectively. $\qquad\square$

First we show that $x_-$ is 1.

**Proposition B.10.** *Let $X_1, \ldots, X_d$ be i.i.d. with $X_i \sim \frac{1}{k}\chi_k^2$, and let $d \geq 2$ be fixed. For any convex combination $\lambda = (\lambda_1, \ldots, \lambda_d)$ with $\lambda_j \geq 0$ and $\sum_{j=1}^d \lambda_j = 1$, define $X(\lambda) = \sum_{j=1}^d \lambda_j X_j$ with CDF $F_\lambda$. Let $e = (1, 0, \cdots, 0)$ denote the vertex. For $\lambda \neq e$, let $x_-(\lambda)$ denote the unique point in $(0, \infty)$ where $F_\lambda(x) = F_e(x)$. Then $x_- = 1$.*

*Proof.* The proof has two parts as follows.
*Part 1: $x_- \geq 1$.*

This primarily follows from Proposition B.12 which shows that the function $g(\lambda) = \mathbb{P}(X(\lambda) \leq 1)$ is Schur-convex. Observe that $\lambda \preceq e$, thus $g(\lambda) \leq g(e)$. Further, from Theorem B.7, we show that the CDFs of $X(e)$ and $X(\lambda)$ cross only once:

$$F\big(t; X(e)\big) \ \geq \ F\big(t; X(\lambda)\big) \quad \text{for } t \leq t^\star, \qquad F\big(t; X(e)\big) \ \leq \ F\big(t; X(\lambda)\big) \quad \text{for } t \geq t^\star.$$

The above two together establish that $x_- \geq 1$.

*Part 2: $x_- \leq 1$:* This follows from item 1 in Proposition 2 in (Székely & Bakirov, 2003) which shows that for all $x \in (1, 2)$

$$\sup_{\lambda \in [0,1]} \mathbb{P}\left(\lambda X_1 + (1-\lambda)X_2 \leq x\right) > \max\left(\mathbb{P}\left(X_1 \leq x\right), \frac{1}{2}(X_1 + X_2) \leq x\right) > \mathbb{P}\left(X_1 \leq x\right).$$

Note that the LHS is a special case of $X(\lambda)$ with only two non-zero entries, and yet it beats $X(e) = X_1$ for all $x \in (1, 2)$. This shows that $x_- \leq 1$ or $x_- \geq 2$. To remove the latter, we use Proposition B.11 which shows that the function $g_x(\lambda) = \mathbb{P}(X(\lambda) \leq x)$ is Schur-concave for $x \geq 2$. This yields $x_- \leq 1$.

Taken together, this establishes that $x_- = 1$.

$\qquad\square$

**Proposition B.11.** *Let $X_1, \ldots, X_d \overset{i.i.d.}{\sim} \frac{1}{k}\chi^2(k)$. For $\lambda \in \Delta^{d-1}$, define*

$$X(\lambda) = \sum_{i=1}^d \lambda_i X_i, \qquad g_t(\lambda) = \mathbb{P}(X(\lambda) \leq t).$$

*For all $t \geq 2$, the function $g_t(\lambda)$ is Schur-concave on $\Delta^{d-1}$.*

*Proof.* This directly follows from Corollary 3 in (Székely & Bakirov, 2003). $\qquad\square$

**Proposition B.12.** *Let $X_1, \ldots, X_d \overset{i.i.d.}{\sim} \frac{1}{k}\chi^2(k)$ with density $f$. For $\lambda \in \Delta^{d-1}$, define*

$$X(\lambda) = \sum_{i=1}^{d} \lambda_i X_i, \qquad g(\lambda) = \mathbb{P}(X(\lambda) \le 1).$$

*Then $g(\lambda)$ is Schur-convex on $\Delta^{d-1}$.*

*Proof of Proposition B.12.* The function $g$ is symmetric (since the $X_i$ are i.i.d.) and continuously differentiable on $\Delta^{d-1}$. By Lemma A.17, it suffices to verify the Schur-Ostrowski condition. By Lemma B.13, this condition for the pair $(i, j)$ is equivalent to $(\lambda_i - \lambda_j) \cdot I(\lambda) \le 0$, where $I(\lambda)$ is the boundary integral of $(x_i - x_j) \prod_k f(x_k)$. By Lemma B.14, it suffices to verify this for $d = 2$. Setting $\lambda_1 = t$ and $\lambda_2 = 1 - t$, the condition becomes $(2t - 1)F(t) \le 0$, which holds by Lemma B.21. Thus $g$ is Schur-convex. $\qquad\square$

**Lemma B.13** (Derivative Formula). *For $\lambda \in \mathrm{int}(\Delta^{d-1})$,*

$$\frac{\partial g}{\partial \lambda_i} - \frac{\partial g}{\partial \lambda_j} = -\frac{1}{\|\lambda\|_2} \int_{\sum_k \lambda_k x_k = 1} (x_i - x_j) \prod_{k=1}^{d} f(x_k)\, dS,$$

*where $dS$ is the surface measure on the hyperplane $\{\sum_k \lambda_k x_k = 1\}$.*

*Proof.* We have $g(\lambda) = \int_{\sum_k \lambda_k x_k \le 1} \prod_k f(x_k)\, dx$. By the Reynolds transport theorem, differentiating with respect to $\lambda_i$ yields a boundary integral. The boundary $\partial\Omega = \{x : \sum_k \lambda_k x_k = 1\}$ has outward unit normal $\hat{n} = \lambda/\|\lambda\|_2$. When $\lambda_i$ increases, the boundary moves with normal velocity $-x_i/\|\lambda\|_2$. Thus:

$$\frac{\partial g}{\partial \lambda_i} = -\frac{1}{\|\lambda\|_2} \int_{\partial\Omega} x_i \prod_k f(x_k)\, dS.$$

The result follows by subtraction.

$\qquad\square$

**Lemma B.14** (Reduction to $d = 2$). *To verify the Schur-Ostrowski condition for $g$, it suffices to consider the case $d = 2$.*

*Proof.* By symmetry of $g$, it suffices to verify the condition for the pair $(1, 2)$. In particular, $d > 2$, the integral in Lemma B.13 involves only $(x_1 - x_2)$. Further, we can decompose the boundary as $\sum_{i>2} \lambda_i x_i = C$ and $\lambda_1 x_1 + \lambda_2 x_2 = 1 - C$. The inner integration with $x_3, \ldots, x_d$ over $\sum_{i>2} \lambda_i x_i = C$ factors out since the $X_k$ are independent giving a positive constant. The resulting structure is identical to $d = 2$. $\qquad\square$

**Definition B.15.** For $d = 2$ and $t \in [0, 1]$, define

$$F(t) := \int_{tx_1 + (1-t)x_2 = 1} (x_1 - x_2)\, f(x_1) f(x_2)\, dS.$$

By Lemmas A.17–B.14, Proposition B.12 reduces to showing:

$$(2t - 1) \cdot F(t) \le 0 \quad \text{for all } t \in [0, 1]. \tag{3}$$

**Lemma B.16** (Anti-symmetry). *$F(t) = -F(1 - t)$ for all $t \in [0, 1]$. In particular, $F(1/2) = 0$.*

*Proof.* Apply the change of variables $(x_1, x_2) \mapsto (y_1, y_2) := (x_2, x_1)$. The constraint $tx_1 + (1 - t)x_2 = 1$ becomes $(1 - t)y_1 + ty_2 = 1$. The integrand $(x_1 - x_2)f(x_1)f(x_2)$ transforms to $(y_2 - y_1)f(y_2)f(y_1) = -(y_1 - y_2)f(y_1)f(y_2)$. The Jacobian is 1. Hence $F(t) = -F(1 - t)$. $\qquad\square$

**Lemma B.17** (Boundary Values). *$F(0) = F(1) = 0$.*

*Proof.* At $t = 0$, the constraint is $x_2 = 1$, giving $dS = dx_1$. Thus:

$$F(0) = \int_0^\infty (x_1 - 1)f(x_1)f(1)\, dx_1 = f(1)\left(\mathbb{E}[X_1] - 1\right) = 0,$$

since $\mathbb{E}[X_1] = 1$. By Lemma B.16, $F(1) = -F(0) = 0$. □

**Lemma B.18** (Derivative at Boundary). $F'(0) = \dfrac{2}{k}f(1) > 0$.

*Proof.* Parametrize the boundary by $x_1 \in [0, 1/t]$ with $x_2 = (1 - tx_1)/(1 - t)$. By Leibniz's rule with moving endpoints, $F'(t) = \int_0^{1/t} \frac{\partial h}{\partial t}\, dx_1$ plus a boundary term that vanishes since $f(0) = 0$ for $k \geq 2$. At $t = 0$: $x_2 = 1$, and $\frac{\partial x_2}{\partial t}\big|_{t=0} = 1 - x_1$. Differentiating the integrand:

$$\left.\frac{\partial h}{\partial t}\right|_{t=0} = (x_1 - 1)f(x_1)f(1) - (x_1 - 1)^2 f(x_1)f'(1).$$

Integrating:

$$F'(0) = f(1) \underbrace{\int_0^\infty (x_1 - 1)f(x_1)\, dx_1}_{=0} - f'(1) \underbrace{\int_0^\infty (x_1 - 1)^2 f(x_1)\, dx_1}_{=2/k}.$$

For $f(x) = Cx^{k/2-1}e^{-kx/2}$, we have $f'(1) = f(1)\left(\frac{k/2-1}{1} - \frac{k}{2}\right) = -f(1)$. Thus:

$$F'(0) = -(-f(1)) \cdot \frac{2}{k} = \frac{2}{k}f(1) > 0. \qquad □$$

**Lemma B.19** (Derivative Symmetry). $F'(t) = F'(1-t)$ *for all* $t \in [0,1]$. *In particular,* $F'(1) = F'(0) > 0$.

*Proof.* Differentiate $F(t) = -F(1-t)$ to obtain $F'(t) = F'(1-t)$. □

**Lemma B.20** (Sign of $F$). $F(t) > 0$ *for* $t \in (0, 1/2)$ *and* $F(t) < 0$ *for* $t \in (1/2, 1)$.

*Proof.* By Lemmas B.17 and B.18, $F(0) = 0$ and $F'(0) > 0$, so $F(t) > 0$ for $t \in (0, \varepsilon)$ for some $\varepsilon > 0$. By Lemma B.16, $F(1/2) = 0$. Since $F(t) = -F(1-t)$ and $F > 0$ near 0, we have $F < 0$ near 1. Combined with $F(1) = 0$ and $F'(1) > 0$ (Lemma B.19), the function $F$ is increasing at $t = 1$, confirming $F(t) < 0$ for $t$ slightly less than 1. Suppose $F(t^*) = 0$ for some $t^* \in (0, 1/2)$. Then $F$ has zeros at $0, t^*, 1/2, 1 - t^*, 1$ (using Lemma B.16). By Rolle's theorem, $F'$ has at least 4 zeros in $(0, 1)$. By Lemma B.19, zeros of $F'$ come in symmetric pairs about $1/2$. But $F'(0) > 0$, $F'(1) > 0$, and $F$ must decrease through $1/2$ (transitioning from positive to negative), so $F'(1/2) < 0$. This forces $F'$ to have a specific structure (positive at endpoints, negative at center) incompatible with 4 or more zeros. Therefore, $F$ has no zeros in $(0, 1/2)$, and $F > 0$ on $(0, 1/2)$. By Lemma B.16, $F(t) = -F(1-t) < 0$ for $t \in (1/2, 1)$. □

**Lemma B.21** (Schur-Ostrowski Condition). *For all* $t \in [0, 1]$: $(2t - 1) \cdot F(t) \leq 0$.

*Proof.* This follows from Lemma B.20:

- If $t \in (0, 1/2)$: $2t - 1 < 0$ and $F(t) > 0$, so $(2t - 1)F(t) < 0$.

- If $t = 1/2$: $2t - 1 = 0$, so $(2t - 1)F(t) = 0$.

- If $t \in (1/2, 1)$: $2t - 1 > 0$ and $F(t) < 0$, so $(2t - 1)F(t) < 0$.

- If $t \in \{0, 1\}$: $F(t) = 0$ by Lemma B.17, so $(2t - 1)F(t) = 0$. □

Now we focus on $x_+$ and establish an asymptotic form of $x_+$.

**Proposition B.22.** *Let $X_1, \ldots, X_d$ be i.i.d. with $X_i \sim \frac{1}{k}\chi_k^2$, and let $d \geq 2$ be fixed. For any convex combination $\lambda = (\lambda_1, \ldots, \lambda_d)$ with $\lambda_j \geq 0$ and $\sum_{j=1}^{d} \lambda_j = 1$, define $X(\lambda) = \sum_{j=1}^{d} \lambda_j X_j$ with CDF $F_\lambda$. Let $u = (1/d, \ldots, 1/d)$ denote uniform weights.*

*For $\lambda \neq u$, let $x_+(\lambda)$ denote the unique point in $(0, \infty)$ where $F_\lambda(x) = F_u(x)$ and $x_+ = \sup_{\lambda \neq u} x_+(\lambda)$. Then*

1. $x_+ \in [1, 2]$

2. $x_+ = 1 + \frac{2}{kd} + O(k^{-2}) \quad as \ k \to \infty.$

*Proof.* Let $g_t(\lambda) = \mathbb{P}\left(X(\lambda) \leq t\right)$. From Proposition B.12 and B.11, we have that the functions $g_1$ and $g_2$ are Schur-convex and Schur-concave respectively. Further, since that $\lambda \succeq u$, thus $g_1(\lambda) \geq g(u)$ and $g_2(\lambda) \leq g(u)$. Finally, from Theorem B.7, we show that the CDFs of $X(u)$ and $X(\lambda)$ cross only once. These together establish that $x_+ \in [1, 2]$.

For the second item, we proceed in four steps: (1) establish the perturbation framework, (2) derive the density difference with explicit error bounds, (3) prove existence and uniqueness of the crossover, and (4) compute its location.

*Step 1: Setup and Characteristic Functions.*

Each $X_i \sim \text{Gamma}(k/2, k/2)$ has mean 1 and variance $2/k$. The characteristic function is

$$\phi(t) = \left(1 - \frac{2it}{k}\right)^{-k/2}.$$

For weighted sums, $\Psi_\lambda(t) = \prod_{j=1}^{d}(1 - 2it\lambda_j/k)^{-k/2}$.

Write $\lambda_j = \frac{1}{d} + \delta_j$ where $\sum_{j=1}^{d} \delta_j = 0$ and $\|\delta\|^2 = \sum_{j=1}^{d} \delta_j^2 > 0$ for $\lambda \neq u$. Note that $\|\delta\|^2 \leq 2$ for any valid $\lambda$.

*Step 2: Cumulant Expansion with Error Bounds.*

Let $\theta = 2it/k$ and $A = 1 - \theta/d$. The cumulant generating function is

$$K_\lambda(t) = -\frac{k}{2} \sum_{j=1}^{d} \ln\left(1 - \theta\lambda_j\right) = -\frac{k}{2} \sum_{j=1}^{d} \left[\ln A + \ln\left(1 - \frac{\theta\delta_j}{A}\right)\right].$$

For $|z| < 1$, we have $\ln(1 - z) = -z - \frac{z^2}{2} - R(z)$ where $|R(z)| \leq \frac{|z|^3}{3(1-|z|)}$.

Setting $z_j = \theta\delta_j/A$, and noting that for $|t| \leq k^{1/2}$ and $k$ sufficiently large, $|z_j| \leq C|\delta_j|/\sqrt{k}$ for some constant $C > 0$, we obtain

$$K_\lambda(t) - K_u(t) = \frac{k\theta^2}{4A^2}\|\delta\|^2 + E_1(t),$$

where the error satisfies $|E_1(t)| \leq C_1\|\delta\|^3|t|^3/k^2$ for $|t| \leq k^{1/2}$.

*Step 3: From Characteristic Functions to Densities.*

Exponentiating and using $|e^w - 1 - w| \leq |w|^2 e^{|w|}$ for the linearization:

$$\Psi_\lambda(t) - \Psi_u(t) = -\frac{\|\delta\|^2}{k} \cdot \frac{t^2\Psi_u(t)}{A^2} + E_2(t),$$

where $|E_2(t)| \leq C_2\|\delta\|^4 t^4 |\Psi_u(t)|/k^2$ for $|t| \leq k^{1/2}$.

The term $\Psi_u(t)/A^2 = (1 - 2it/(kd))^{-(kd/2+2)}$ is the CF of $g \sim \text{Gamma}(\alpha, \beta)$ with $\alpha = kd/2 + 2$ and $\beta = kd/2$.

By Fourier inversion, since both $f_\lambda$ and $f_u$ are smooth densities with all moments finite, and the CF difference is integrable:

$$f_\lambda(x) - f_u(x) = \frac{\|\delta\|^2}{k}g''(x) + r(x),$$

where $\|r\|_\infty = O(\|\delta\|^3/k^2)$ uniformly over $x > 0$.

*Step 4: Existence, Uniqueness, and Location of Crossover.*

**Lemma B.23.** *For $k$ sufficiently large and any $\lambda \neq u$, there exists a unique $x_+(\lambda) \in (0,\infty)$ with $F_\lambda(x_+) = F_u(x_+)$.*

*Proof of Lemma.* Define $\Delta(x) = F_\lambda(x) - F_u(x) = \int_0^x (f_\lambda - f_u)\,dy$. Using Step 3:

$$\Delta(x) = \frac{\|\delta\|^2}{k}\big[g'(x) - g'(0)\big] + O(\|\delta\|^3/k^2).$$

Since $\alpha = kd/2 + 2 > 2$, we have $g'(0) = 0$. Thus $\Delta(x) = \frac{\|\delta\|^2}{k}g'(x) + O(\|\delta\|^3/k^2)$.

The function $g'(x)$ is strictly positive on $(0, x_*)$ and strictly negative on $(x_*, \infty)$, where $x_* = (\alpha - 1)/\beta = 1 + 2/(kd)$ is the mode. Since $\|\delta\|^2 \geq c > 0$ for $\lambda$ bounded away from $u$, the leading term dominates for large $k$, ensuring exactly one sign change. $\qquad\square$

The crossover occurs where $\Delta(x_+) = 0$. Setting $\frac{\|\delta\|^2}{k}g'(x_+) + O(\|\delta\|^3/k^2) = 0$ and using implicit function theorem:

$$x_+ = x_* + O(k^{-2}) = 1 + \frac{2}{kd} + O(k^{-2}).$$

Since this expression is independent of $\delta$ at leading order, and the $O(k^{-2})$ error is uniform over $\|\delta\|^2 \in (0, 2]$:

$$\sup_{\lambda \neq u} x_+(\lambda) = 1 + \frac{2}{kd} + O(k^{-2}). \qquad\square$$

### B.6.1. SIMULATIONS

We run simulations to show that $x_-$ is essentially 1 and the asymptotic form of $x_+$.

We start with $x_-$. Let $e = (1, 0, \cdots, 0)$ and $u = (d^{-1}, d^{-1}, \cdots, d^{-1})$ be vertex and center respectively. Consider the path $\lambda(t) = te + (1 - t)u$.

We run simulations to show that for every $\epsilon > 0$, there exists $t > 0$ such that $F(1 + \epsilon; X(\lambda(t))) > F(1 + \epsilon; X(e))$. This establishes that $x_- 1$.

**Procedure.** We fix a small $\epsilon$, and do a binary search over $t$ to find the $t$ such that $F(1 + \epsilon; X(\lambda(t))) > F(1 + \epsilon; X(e))$. We also compute numerical derivatives at $1 + \epsilon$ with $t = 0$. We report observations below.

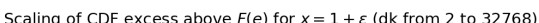

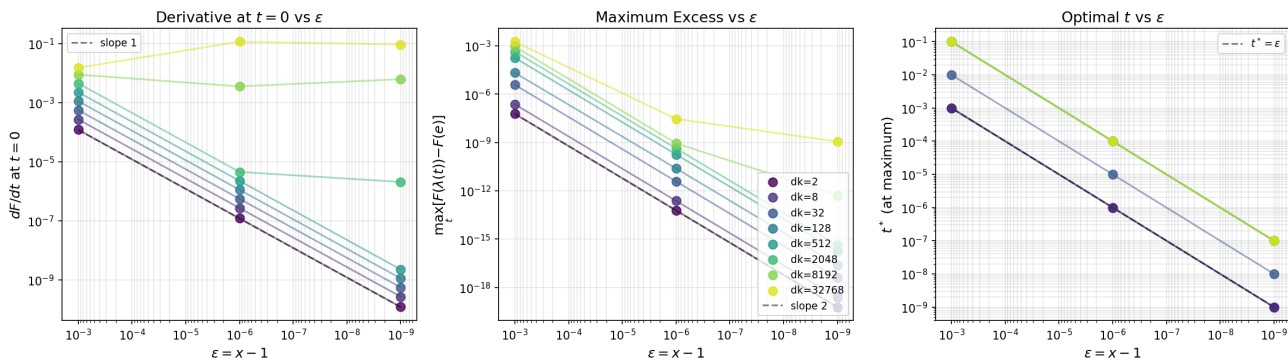

*Figure 6.* Simulations showing $x_-$ is *essentially* 1

- **Left Panel: Derivative at** $t = 0$ **vs** $\epsilon$, Shows $\frac{\partial F}{\partial t}|_{x=1+\epsilon, t=0}$. Rate of change of the CDF as we move away from the vertex. We see All lines have slope 1 on log-log, meaning derivative $\propto \epsilon$. This means that he CDF immediately starts increasing when you move away from vertex, and this effect gets stronger as $dk$ increase

- **Middle Panel: Maximum Excess vs** $\epsilon$. Shows $\max_t[F(1+\epsilon; X(\lambda(t))) - F(1+\epsilon; X(e))]$, the maximum amount the CDF exceeds the vertex value. All lines have slope 2 on log-log, meaning max diff $\propto \epsilon^2$. For larger $dk$, the improvement is more dramatic

- **Right Panel: Optimal** $t$ **vs** $\epsilon$. Shows $t^*$ where the maximum excess occurs. Lines have slope 1, meaning $t^* \propto \epsilon$

We now turn to $x_+$. We simulate the computation of $x_+$ for $k \in \{1, 2, 4, \cdots 32\}$ and $d \in \{2, 4, 8, \cdots 2048\}$. For each $(k, d)$, we find $x_+$ via binary search. Consider a black-box routine which gives the solution of the following.

$$i, j, \lambda = \underset{i,j \in \mathbb{N}, \lambda: 1 \le i+j \le d, \lambda \in [0, 0.5]}{\arg\sup} F\left(x; \frac{\lambda}{ik}\chi^2(ik) + \frac{(1-\lambda)}{jk}\chi^2(jk)\right)$$

Starting with $x_{\text{low}} = 1$ and $x_{\text{high}} = 2$, we call the above on the mid-point get $(i, j, \lambda)$. We check if $i = j = \frac{d}{2}$ and $\lambda \approx 0.5$ upto tolerance $\tau$. If yes, we stop, else, we compute $F(x; X(u))$, cdf with uniform distribution. We check if this cdf is larger or smaller, and accordingly update $x_{\text{low}}$ or $x_{\text{high}}$. This gives us a routine to compute $x_+$ in $O(\log(\tau^{-1}))$ calls.

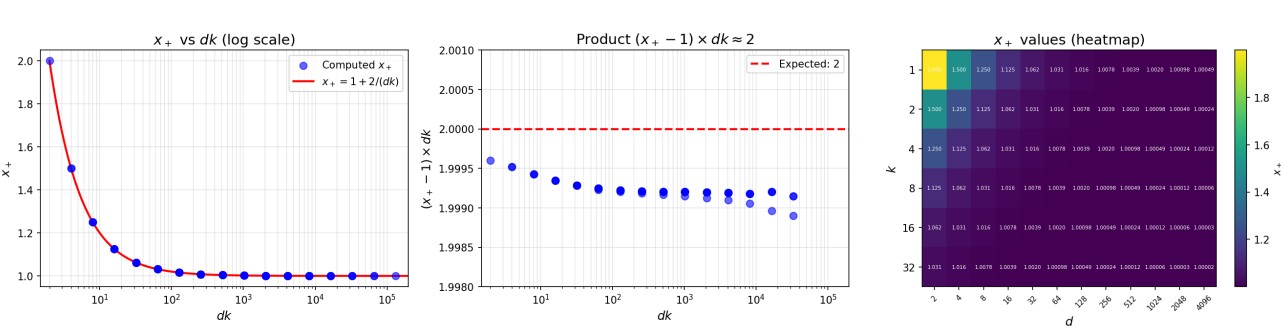

*Figure 7.* Simulations on $x_+$ across settings of $d$ and $k$ together with the $1 + \frac{2}{dk}$ fit.

Figure 7 shows the critical threshold $x_+$ where the uniform allocation ($\lambda = 0.5, i = j = d/2$) becomes optimal for maximizing the CDF of the weighted chi-squared mixture.

- **Left panel:** $x_+$ **vs** $dk$ **(log scale):** Scatter plot of numerically computed $x_+$ values against the product $dk$, with the theoretical curve $x_+ = 1 + \frac{2}{dk}$ overlaid in red. The close agreement across 72 data points spanning $dk \in [2, 131072]$ confirms the analytical formula. As $dk$ increases, $x_+$ approaches 1 from above, indicating that uniform allocation becomes optimal for increasingly smaller deviations from $x = 1$.

- **Middle panel: Verification of** $(x_+ - 1) \times dk \approx 2$**:** The product $(x_+ - 1) \times dk$ is plotted to verify the scaling relationship. All computed values cluster tightly around 2 (dashed red line), with deviations less than $0.5\%$, confirming that $x_+ - 1$ scales as $2/(dk)$.

- **Right panel: Heatmap of** $x_+$ **values:** $x_+$ for different $(k, d)$ combinations. The color gradient from yellow ($x_+ \approx 2$) to purple ($x_+ \approx 1$) illustrates that $x_+$ depends only on the product $dk$: cells with the same $dk$ value share identical $x_+$ regardless of individual $k$ and $d$ values. This confirms that the transition point is governed by the effective dimension $dk$ rather than $k$ or $d$ separately.

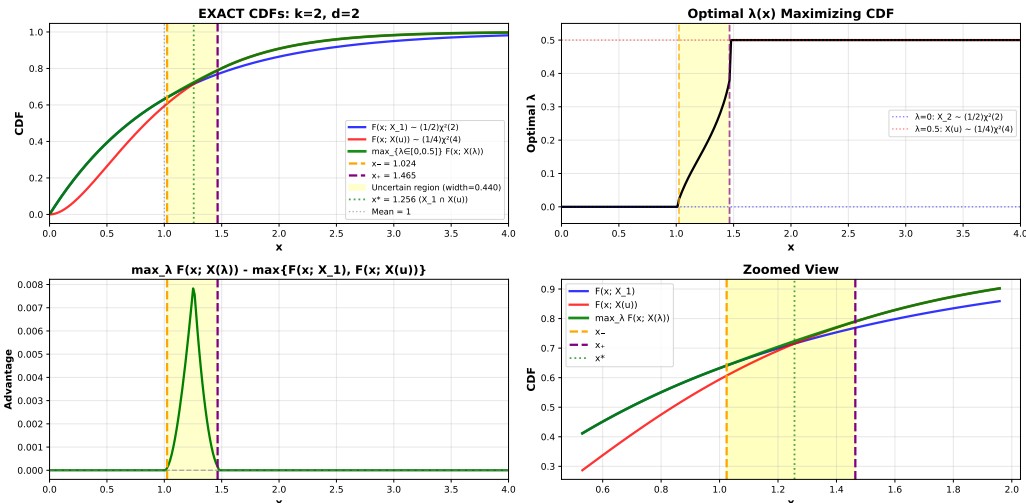

*Figure 8.* Simulations of the envelope CDF for $k = 2, d = 2$

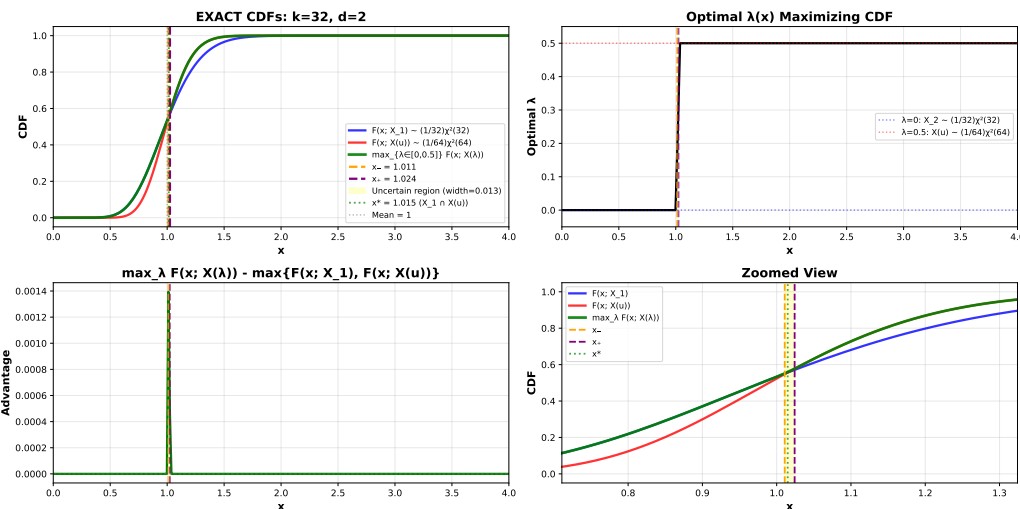

*Figure 9.* Simulations of the envelope CDF for $k = 32, d = 2$

## B.7. Proof of Main Theorem 5.1

Let $D$ and $D'$ be neighboring datasets with differing element $(A_0, G_0)$ and let $Q_0 := A^\top G$. We start with Lemma B.2, which shows that for any norm estimation routine, $x \mapsto R(x)$, the trade-off function is bounded as,

$$T(\mathcal{A}(D)\|\mathcal{A}(D')) \succeq T\left(Z, \mathcal{N}(Z, 1)\|Z, \mathcal{N}(0, 1)\right)$$

where $Z = \frac{\|Q_0\|}{R(Q_0)}$. We now instantiate $R$ with Hutch and Hutch$^{++}$ and specialize the analysis.

### B.7.1. HUTCH.

Recall that with $P \in \mathbb{R}^{k \times p}$, $A \in \mathbb{R}^{T \times d}$ and $G \in \mathbb{R}^{T \times p}$, the Hutch estimator is,

$$\left\| P(A^\top G)^\top \right\|_F^2 = \text{trace}(P^\top (A^\top G)(A^\top G)^\top P) = \text{trace}(P^\top OP) = \sum_{i=1}^{k} P_i^\top OP_i$$

where $O = (A^\top G)(A^\top G)^\top$. We restrict to the differing element, $(A_0, G_0)$ which us $Q_0 = A_0^\top G_0$ and correspondingly, $O_0$. Observe that $O_0$ is a $d \times d$ positive semi-definite matrix. Hence, by diagonalization of symmetric positive semi-definite matrices, we can write down $O_0 = U\Sigma U^\top$ where $\Sigma = diag(\lambda_1, \cdots, \lambda_d)$. Substituting this into the summand for $P_i \sim \mathcal{N}(0, \frac{1}{k} I_d)$, we get that

$$P_i^\top O_0 P_i = P_i^\top U\Sigma U^\top P_i$$

Since $U$ is an orthonormal matrix and Gaussians are rotationally invariant, we get that $Z = U^\top P_i \sim \mathcal{N}(0, \frac{1}{k} I_d)$. Thus, we get that

$$P_i^\top O_0 P_i = \sum_{j=1}^{d} \lambda_j Z_{jj}^2 \sim \sum_{j=1}^{d} \frac{\lambda_j}{k} \chi_j^2(1)$$

For the above, the distribution of summand, $P_i^\top O_0 P_i$ is $\sum_{j=1}^{d} \lambda_j \chi_j(1)^2$ where $\lambda_j$ are eigenvalues of $O_0$. Further, the distribution of the sum, $\sum_{i=1}^{k} P_i^\top O P_i$ is the generalized chi-squared distribution $\sum_j \lambda_j \chi_j(k)^2$ with weights $\{\lambda_j\}$. This follows since each $P_i$ is independent of each other for $i \in [k]$, the distribution of the sum $\sum_{i=1}^{k} P_i^\top O_0 P_i$ becomes a generalized chi-squared distribution $\sum_{j=1}^{d} \frac{\lambda_j}{k} \chi_j^2(k)$ due to the definition of $\chi^2(k)$ distribution to be sum of $k$ independent $\chi^2(1)$ distributions.

Finally,

$$\|O_0\|_F^2 = \text{trace}((A_0^\top G_0)(A_0^\top G_0)^\top) = \text{trace}(O_0) = \sum_{i=1}^{k} \lambda_i = \|\lambda\|_1$$

Combining we get,

$$Z^2 \sim \frac{\|\lambda\|_1}{\sum_i \lambda_i \chi^2(k)}$$

The above is still data-dependent (though $\lambda$) – we denote it as $Z(\lambda)$ from now. Note that $Z(\lambda)$ is scale-invariant, so we restrict to $\|\lambda\|_1 = 1$, giving us $Z(\lambda) \sim (\sum_i \lambda_i \chi^2(k))^{-1}$.

Proposition 5.2 gives us a data-indepedent random variable $Y$ such that $Y \preceq_{st} \sum_i \lambda_i \chi^2(k) \implies Y^{-1} \succeq_{st} Z(\lambda)$ for all $\lambda$ with $\|\lambda\|_1 = 1$

We finally apply Lemma B.3 by with $Y^{-1} \succeq_{st} Z(\lambda)$ to get

$$T(\mathcal{A}(D)\|\mathcal{A}(D')) \succeq T(Z, \mathcal{N}(Z, 1)\|Z, \mathcal{N}(0, 1)) \succeq T(Y^{-1}, \mathcal{N}(Y^{-1}, 1)\|Y^{-1}, \mathcal{N}(0, 1)).$$

### B.7.2. HUTCH$^{++}$

We recall the Hutch$^{++}$ estimator,

$$\mathsf{Hutch}_k^{++}(O) = \left\|(QG^\top)A\right\|_F^2 + \left\|(P((I - QQ^\top)A^\top G)^\top\right\|_F^2$$

where $P \in \mathbb{R}^{p \times k}$ consists of i.i.d $\mathcal{N}(0, 1/k)$ entries as before, and $Q \in \mathbb{R}^{p \times k}$ is the orthogonal basis for the span of $OS = A(G^\top S)$ where $S$ also consists of i.i.d $\mathcal{N}(0, 1/k)$ entries.

For simplified privacy analysis, we consider the following stronger adversary: the adversary, in addition to knowing $A_0, G_0$, knows $Q_0$.

Let $\{\lambda_i\}_{i=1}^{k}$ and $\{\lambda_i\}_{i=k+1}^{d}$ denote the (non-zero) eigenvalues of $Q_0^\top O_0$ and $(I - Q_0 Q_0^\top)^\top O_0 (I - Q_0 Q_0^\top)$ respectively. We thus have,

$$\mathsf{Hutch}_k^{++}(O) \sim \sum_{i=1}^{k} \lambda_i + \frac{1}{k} \sum_{i=k+1}^{d} \lambda_i \chi_j^2(k)$$

where we used the adversary assumption to make the first summand deterministic (yet worst-case).

Since $Q_0$ is orthogonal, from Pythagoras theorem,

$$\|U_o\|_F^2 = \|Q_0 U_0\|_F^2 + \left\|(I - Q_0 Q_0^\top) U_0\right\|_F^2$$
$$= \operatorname{trace}(Q_0^\top O_0 Q_0) + \operatorname{trace}((I - QQ^\top)^\top O(I - QQ^\top))$$
$$= \sum_{i=1}^d \lambda_i = \|\lambda\|_1$$

As before, combining we get,

$$Z^2 \sim \frac{\|\lambda\|_1}{\sum_{i=1}^k \lambda_i + \frac{1}{k} \sum_{i=k+1}^d \lambda_i \chi_j^2(k)}$$

The above is still data-dependent (though $\lambda$) – we denote it as $Z(\lambda)$ from now. Note that $Z(\lambda)$ is scale-invariant, so we restrict to $\|\lambda\|_1 = 1$, giving us $Z(\lambda) \sim (\sum_{i=1}^k \lambda_i + \frac{1}{k} \sum_{i=k+1}^d \lambda_i \chi_j^2(k))^{-1}$.

Finally, Proposition B.24 gives us a data-interdependent random variable $Y$ such that $Y \preceq_{st} \sum_i \lambda_i \chi^2(k) \implies Y^{-1} \succeq_{st} Z(\lambda)$ for all $\lambda$ with $\|\lambda\|_1 = 1$

We finally apply Lemma B.3 by with $Y^{-1} \succeq_{st} Z(\lambda)$ to get

$$T(\mathcal{A}(D)\|\mathcal{A}(D')) \succeq T\left(Z, \mathcal{N}(Z,1)\|Z, \mathcal{N}(0,1)\right) \succeq T\left(Y^{-1}, \mathcal{N}(Y^{-1},1)\|Y^{-1}, \mathcal{N}(0,1)\right).$$

## B.8. Results and Proofs for Hutch$^{++}$

**Proposition B.24** (Envelope for mixtures of deterministic ones and Chi-squared). *Fix integers $d \geq k \geq 1$. Let $X_{k+1}, \ldots, X_d$ be i.i.d. with $X_i \sim \frac{1}{k}\chi^2(k)$ (so $\mathbb{E}[X_i] = 1$), and set $X_i \equiv 1$ for $i = 1, \ldots, k$. For $\lambda \in \Delta^{d-1} := \{\lambda \in \mathbb{R}_{\geq 0}^d : \sum_{i=1}^d \lambda_i = 1\}$, define the mixture*

$$X(\lambda) := \sum_{i=1}^d \lambda_i X_i, \qquad f(x) := \sup_{\lambda \in \Delta^{d-1}} F(x; X(\lambda)) = \sup_{\lambda \in \Delta^{d-1}} \Pr\left[X(\lambda) \leq x\right].$$

*Then:*

1. *For all $x < 1$, $f(x) = F(x; X)$, where $X \sim \frac{1}{k}\chi^2(k)$*

2. *For all $x \geq 1$, $f(x) = 1$.*

*Proof.* Define $\alpha := \sum_{i \leq k} \lambda_i$ and, if $\alpha < 1$, set $\beta_i := \lambda_i/(1-\alpha)$ for $i > k$ (so $\beta \in \Delta^{d-k-1}$), yielding

$$X(\lambda) = \alpha + (1-\alpha) Z(\beta), \qquad Z(\beta) := \sum_{i>k} \beta_i X_i, \quad \mathbb{E}[Z(\beta)] = 1.$$

From Lemma B.25, we have that for the envelope cdf, for $x < 1$, $\alpha = 0$ and for $x \geq 1$, $\alpha = 1$. This establishes item 2 in the claim.

For item 1, note that $X(\lambda) = \sum_{i=k+1}^d \beta_i X_i$ with $\beta \in \Delta^{d-k-1}$. This is identical to the Hutch case (with the restriction that $x < 1$). Using Proposition 5.2, we have that there exists $x_- \downarrow 1$ (as $k$ or $d \to \infty$), such that for $x \leq x_-$, the extremal map $x \mapsto F(x; X_1)$ is the envelope. Using $x_- > 1$ establishes item 1 and completes the proof. $\qquad\square$

**Lemma B.25.** *Let $X_i \overset{iid}{\sim} \frac{1}{k}\chi^2(k)$ (mean 1) for $i = k+1, \ldots, d$, and let $Y_i := 1$ for $i \leq k$ and $Y_i := X_i$ for $i > k$. For $\lambda \in \Delta^{d-1}$, define*

$$X(\lambda) := \sum_{i=1}^d \lambda_i Y_i, \qquad f(x) := \sup_{\lambda \in \Delta^{d-1}} \Pr\left[X(\lambda) \leq x\right].$$

*Write $\alpha := \sum_{i \leq k} \lambda_i$ and, if $\alpha < 1$, set $\beta_i := \lambda_i/(1-\alpha)$ for $i > k$ (so $\beta \in \Delta^{d-k-1}$), yielding*

$$X(\lambda) = \alpha + (1-\alpha)\,Z(\beta), \qquad Z(\beta) := \sum_{i>k} \beta_i X_i, \quad \mathbb{E}[Z(\beta)] = 1.$$

*Then the maximizing choice of $\alpha$ in the envelope satisfies*

$$\alpha^*(x) = \begin{cases} 0, & x < 1, \\ 1, & x \geq 1. \end{cases}$$

*Proof.* Fix $x \in \mathbb{R}$ and a weight vector $\beta \in \Delta^{d-k-1}$ (when $\alpha < 1$). For $\alpha \in [0,1)$,

$$\Pr\left[X(\lambda) \leq x\right] = \Pr\left[Z(\beta) \leq \frac{x-\alpha}{1-\alpha}\right] = F_{Z(\beta)}\Big(t(\alpha)\Big), \quad t(\alpha) := \frac{x-\alpha}{1-\alpha}.$$

Differentiate $t(\alpha)$:

$$t'(\alpha) = \frac{x-1}{(1-\alpha)^2}.$$

Since $F_{Z(\beta)}$ is non-decreasing (strictly increasing on the support for nondegenerate $Z(\beta)$):

- If $x < 1$, then $t'(\alpha) < 0$, so $\alpha \mapsto F_{Z(\beta)}(t(\alpha))$ is strictly decreasing on $[0,1)$. Hence the supremum over $\alpha$ is attained at $\alpha = 0$, yielding $\alpha^*(x) = 0$.

- If $x \geq 1$, consider $\alpha = 1$. Then $X(\lambda) \equiv 1$ deterministically, so $\Pr[X(\lambda) \leq x] = 1$. For any $\alpha < 1$, we have $t(\alpha) \geq 1$, and for the continuous Gamma law of $Z(\beta)$, $F_{Z(\beta)}(t) < 1$ for any finite $t$. Therefore no choice with $\alpha < 1$ attains probability 1, and the supremum is achieved at $\alpha = 1$, i.e., $\alpha^*(x) = 1$.

Combining the two cases gives the stated optimizer. $\qquad\square$

## B.9. Additional details to Privacy Accounting

In this section, we add additional details related to envelope CDF computation. and privacy accounting with envelope CDF.

### B.9.1. ENVELOPE CDF COMPUTATION

In this section, we give an efficient algorithm for computing the envelope with black-box calls to

$$(i,j,\lambda) = \text{MIDDLEREGIONCDF}(x,k,d) = \arg \max_{\substack{i,j \in \mathbb{N}, \, \lambda \in [0,0.5] \\ 1 \leq i+j \leq d}} F\left(x; \frac{\lambda}{ik}\chi^2(ik) + \frac{(1-\lambda)}{jk}\chi^2(jk)\right)$$

Each call can be implemented in $T_{\text{MR}}(d, n_\lambda) = O(d^2 n_\lambda)$ where $n_\lambda$ denote the size of discreteized $\lambda \in [0,1]$.

Algorithm 4 is computed the envelope CDF given r range $[x_{\min}, x_{\max}]$ and discretization size $\Delta x$. It directly computes CDF for extremal parts: $x \leq 1$ and $x \geq x_+$, and use the blackbox calls to MIDDLEREGIONCDF for the middle-region. Algorithm 4 evaluates $n_{\Delta x} + 1$ grid points; if $m$ points lie in $(1, x_+)$, the cost is $O(m \cdot T_{\text{BB}}(k,d) + (n_{\Delta x} - m))$, since tail evaluations are $O(1)$.

Algorithm 3 computes $x_+$ via binary search using the single-crossing property (which implies monotonicity), Lemma B.4. It operates on the interval $[x_{\text{low}}, x_{\text{high}}] = [1,2]$. At each iteration, the interval is halved. To achieve precision $\tau$, the number of iterations is $T_{\text{iter}} = O\left(\log\left(\frac{1}{\tau}\right)\right) = O(\log\left(\tau^{-1}\right))$. Each iteration requires one call to the black-box optimization routine and one CDF evaluation under the uniform distribution. The total complexity is: $O\left(\log\left(\tau^{-1}\right) \cdot T_{\text{MR}}(d, n_\lambda)\right)$

---

**Algorithm 3** Binary Search for $x_+$

---

**Input:** Dimension $d$, parameter $k$, tolerance $\tau > 0$
**Output:** Approximate value of $x_+$
1: $x_{\text{low}} \leftarrow 1$
2: $x_{\text{high}} \leftarrow 2$
3: **while** $x_{\text{high}} - x_{\text{low}} > \tau$ **do**
4:      $x_{\text{mid}} \leftarrow \frac{x_{\text{low}} + x_{\text{high}}}{2}$
5:      $(i^\star, j^\star, \lambda^\star) \leftarrow \text{MIDDLEREGIONCDF}(x_{\text{mid}}, k, d)$     // Solve $\underset{\substack{i,j \in \mathbb{N},\, \lambda \in [0, 0.5] \\ 1 \leq i+j \leq d}}{\arg\max} F\left(x_{\text{mid}}; \frac{\lambda}{ik}\chi^2(ik) + \frac{(1-\lambda)}{jk}\chi^2(jk)\right)$

6:      **if** $i^\star = j^\star = \frac{d}{2}$ **and** $|\lambda^\star - 0.5| \leq \tau$ **then**
**Output:**      $x_{\text{mid}}$                                                               //Found $x_+$
7:      **end if**
8:      $F_{\text{unif}} \leftarrow F(x_{\text{mid}}; X(u))$                                 // CDF under uniform distribution
9:      **if** $F_{\text{unif}} > 0.5$ **then**
10:        $x_{\text{high}} \leftarrow x_{\text{mid}}$                                           //$x_{\text{mid}}$ is too large
11:      **else**
12:        $x_{\text{low}} \leftarrow x_{\text{mid}}$                                          // $x_{\text{mid}}$ is too small
13:      **end if**
14: **end while**
**Output:** $\frac{x_{\text{low}} + x_{\text{high}}}{2}$

---

**Algorithm 4** Envelope CDF computation

---

**Input:** Dimension $d$, parameter $k$, threshold $x_+$, grid size $n_{\Delta x}$, thresholds bounds $[x_{\min}, x_{\max}]$
**Output:** Array of CDF values $\{(x_i, F_i)\}_{i=0}^{n_{\Delta x}}$
1: $\mathcal{G} \leftarrow \emptyset$                                                        // Initialize output grid
2: $\Delta x \leftarrow (x_{\max} - x_{\min})/n_{\Delta x}$                                 //Compute step size
3: **for** $i = 0$ to $n_{\Delta x}$ **do**
4:      $x_i \leftarrow x_{\min} + i \cdot \Delta x$
5:      **if** $x_i \leq 1$ **then**
6:        $F_i \leftarrow F\left(x_i; \frac{1}{k}\chi^2(k)\right)$                       //Left tail: single $\chi^2(k)$
7:      **else if** $x_i \geq x_+$ **then**
8:        $F_i \leftarrow F\left(x_i; \frac{1}{kd}\chi^2(kd)\right)$                   //Right tail: single $\chi^2(kd)$
9:      **else**
10:        $(i^\star, j^\star, \lambda^\star) \leftarrow \text{MIDDLEREGIONCDF}(x_i, k, d)$
11:        $F_i \leftarrow F\left(x_i; \frac{\lambda^\star}{i^\star k}\chi^2(i^\star k) + \frac{(1-\lambda^\star)}{j^\star k}\chi^2(j^\star k)\right)$     //Middle: mixture
12:      **end if**
13:      $\mathcal{G} \leftarrow \mathcal{G} \cup \{(x_i, F_i)\}$
14: **end for**
**Output:** $\mathcal{G}$

---

### B.9.2. PRIVACY ACCOUNTING

Algorithm 5 describes the privacy accounting procedure. It takes as input the noise multiplier $\sigma$, dataset size $N$, batch size $B$, number of epochs $E$, target delta $\delta_{\text{tgt}}$, mesh size $h$, support cap $t_{\max}$, and optionally an envelope CDF $F(\cdot; Y)$ for randomized clipping. It outputs $\varepsilon^\star$ satisfying $\delta(\varepsilon^\star) = \delta_{\text{tgt}}$ along with the composed privacy-loss distribution.

We use the standard normal CDF $\Phi(\cdot)$ and survival function $\bar{\Phi}(x) = 1 - \Phi(x)$. For randomized clipping, the effective scale variable is $a = 1/\sqrt{Y}$, where $Y$ follows the envelope CDF $F(\cdot; Y)$. Further, the sampling probability is $p = B/N$ with $T = \lceil N/B \rceil$ steps per epoch.

In Algorithm 5, we first construct a symmetric grid of cut points with step $h$ up to $t_{\max}$, deriving cell-centered grid points.

Next, we compute mechanism kernels $\alpha(t)$ and $\beta(t)$: for standard DP-SGD with fixed scale $a = 1$, these are Gaussian CDFs evaluated at shifted arguments; for randomized clipping, they become weighted sums over discretized scale values drawn from $F(\cdot; Y)$.

$$\alpha(t) = \mathbb{E}_Y\Big[\Phi\Big(-\frac{t}{(1/\sqrt{Y})/\sigma} - \frac{(1/\sqrt{Y})/\sigma}{2}\Big)\Big] = \int \Phi\Big(-\frac{t}{a/\sigma} - \frac{a/\sigma}{2}\Big)\, dF(a; A),$$

$$\beta(t) = \mathbb{E}_Y\Big[\Phi\Big(\frac{t}{(1/\sqrt{Y})/\sigma} - \frac{(1/\sqrt{Y})/\sigma}{2}\Big)\Big] = \int \Phi\Big(\frac{t}{a/\sigma} - \frac{a/\sigma}{2}\Big)\, dF(a; A),$$

We apply privacy amplification by subsampling using the transformation $s(t) = \log\left(\frac{1}{p} - \frac{1-p}{p}e^{-t}\right)$ to obtain the single-iteration survival function $S(t)$. We then convert $S(t)$ to per-cell probability masses, clip negatives, and normalize to get $q^{(1)}$. We compose via fast convolution: convolve $q^{(1)}$ with itself $T$ times for one epoch, then $E$ times for the full run, cleaning and normalizing after each step. We evaluate $\delta(\varepsilon)$ by summing tail probabilities weighted by $(1 - e^{\varepsilon-t})$. Finally, we solve for $\varepsilon^\star$ via bracketing and root-finding (e.g., Brent's method), extending $t_{\max}$ if needed.

We use a mesh size $h \approx 10^{-4}$ which balances accuracy and speed. We ensure $F(\cdot; Y)$'s domain captures sufficient mass. We normalize and clip small negatives after numerical operations, and extend $t_{\max}$ if $\delta_{\mathrm{tgt}}$ lies beyond the current support.

---

**Algorithm 5** Privacy Accountant with Envelope CDF

---

**Input:** noise $\sigma$, samples $N$, batch $B$, epochs $E$, target $\delta_{\mathrm{tgt}}$, mesh $h$, max support $t_{\max}$, optional envelope CDF $F(\cdot; Y)$
**Output:** $\varepsilon^\star$ with $\delta(\varepsilon^\star) = \delta_{\mathrm{tgt}}$
1: $T \leftarrow \lceil N/B \rceil, p \leftarrow B/N$
2: Build grids: $t_+^{\mathrm{cdf}} \leftarrow \{h/2, 3h/2, \ldots, t_{\max}\}$; $t^{\mathrm{cdf}} \leftarrow \{-t_+^{\mathrm{cdf}}\} \cup t_+^{\mathrm{cdf}}$; cell-centered $\mathcal{T}$ from midpoints (include 0)
3: **if** $F_Y$ given **then**
4:     Choose $[y_{\min}, y_{\max}]$, grid $y_0 < \cdots < y_M$, set $F_i \leftarrow F(y_i; Y)$ (monotone, clipped $[0, 1]$)
5:     $w_i \leftarrow \max(F_i - F_{i-1}, 0)$, normalize $\sum_i w_i = 1$; $\bar{y}_i \leftarrow (y_{i-1} + y_i)/2$, $a_i \leftarrow 1/\sqrt{\bar{y}_i}$
6:     $\alpha(t) \leftarrow \sum_i w_i\, \Phi\big(-t/(a_i/\sigma) - (a_i/\sigma)/2\big)$; $\beta(t) \leftarrow \sum_i w_i\, \Phi\big(t/(a_i/\sigma) - (a_i/\sigma)/2\big)$
7:     $\overline{\alpha}(t) \leftarrow \sum_i w_i\, \bar{\Phi}\big(-t/(a_i/\sigma) - (a_i/\sigma)/2\big)$; $\overline{\beta}(t) \leftarrow \sum_i w_i\, \bar{\Phi}\big(t/(a_i/\sigma) - (a_i/\sigma)/2\big)$
8: **else**
9:     $\alpha(t) \leftarrow \Phi\big(-t/(1/\sigma) - (1/\sigma)/2\big)$; $\beta(t) \leftarrow \Phi\big(t/(1/\sigma) - (1/\sigma)/2\big)$; define $\overline{\alpha}, \overline{\beta}$ with $\bar{\Phi}(\cdot)$
10: **end if**
11: $t_{\min} \leftarrow \log(1 - p)$ (lower support bound of subsampled PLD)
12: **for** $t \in t^{\mathrm{cdf}}$ **do**
13:     **if** $t \leq t_{\min}$ **then**
14:         $S(t) \leftarrow 0$ (CDF is zero below support)
15:     **else**
16:         $s(t) \leftarrow \log\left(\frac{1}{p} - \frac{1-p}{p}e^{-t}\right)$
17:         $S(t) \leftarrow p\,\overline{\beta}(t+s(t)) + (1-p)\,\alpha(t+s(t))$
18:     **end if**
19: **end for**
20: Build single-iteration PDF $q^{(1)}$ on $\mathcal{T}$ by differences of $S$ across cells; clip $< 0$; normalize
21: $q^{\mathrm{epoch}} \leftarrow \mathrm{FastConvolve}(q^{(1)}, T)$; $q^{\mathrm{total}} \leftarrow \mathrm{FastConvolve}(q^{\mathrm{epoch}}, E)$
22: Define $\delta(\varepsilon)$: find first $i$ with $t_i \in \mathcal{T}$ and $t_i \geq \varepsilon$; $\delta(\varepsilon) \leftarrow \sum_{j \geq i} q_j^{\mathrm{total}}\big(1 - e^{\varepsilon - t_j}\big)$, clipped to $[0, 1]$
23: Find bracket in $[0, t_{\max}]$ and solve $\delta(\varepsilon) = \delta_{\mathrm{tgt}}$ (1D root-find) $\rightarrow \varepsilon^\star$
**Output:** $\varepsilon^\star$

---

### B.10. Proof structure

We give a high-level description of the proof structure below.

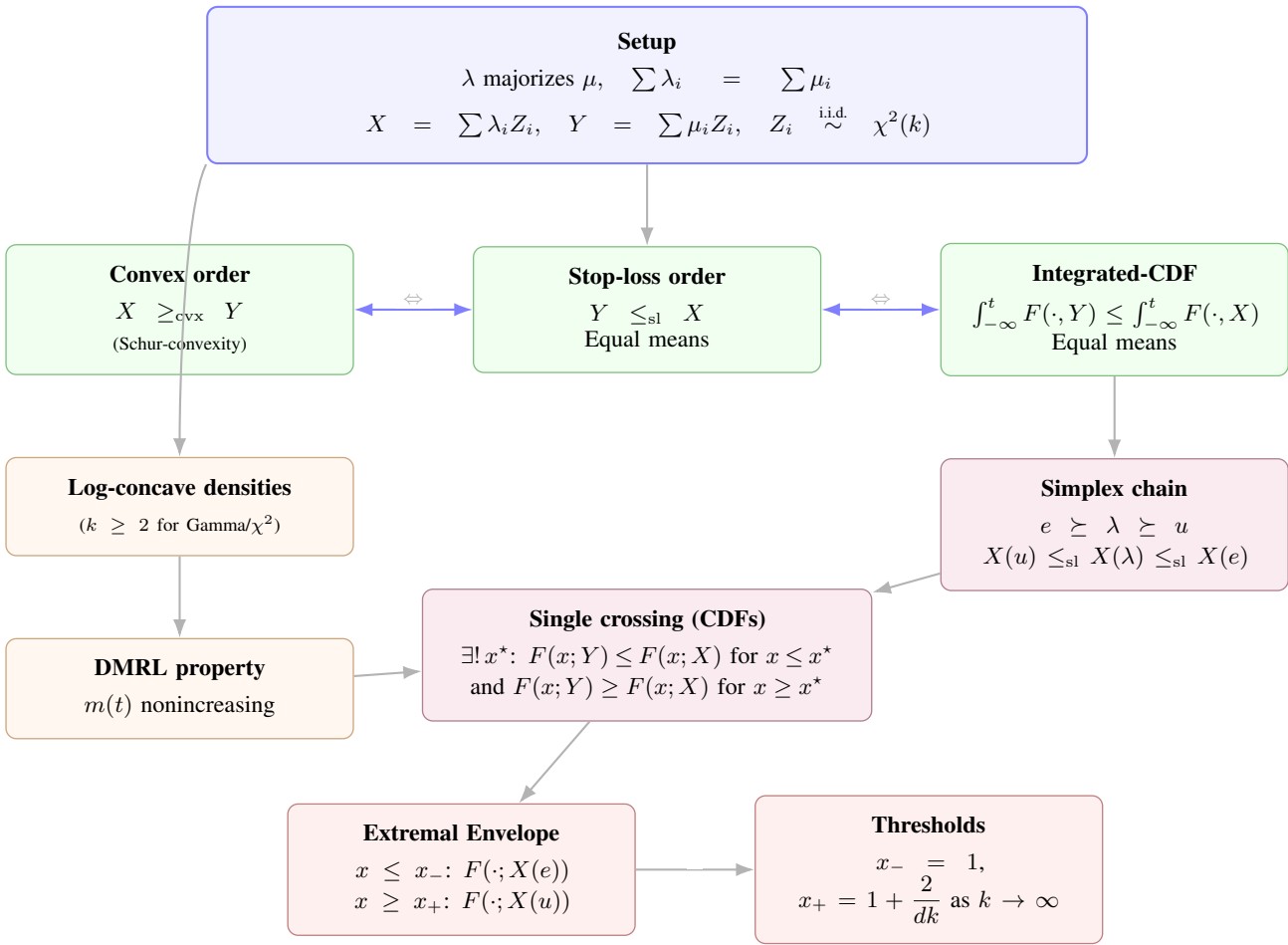

*Figure 10.* A high-level structure proof of Proposition 5.2

For convenience, we summarize the principal notation used in the analysis in Table 5.

| Symbol | Definition |
|---|---|
| $F(\cdot; X)$ | CDF of random variable $X$ |
| $Z$ | Ratio of estimated per-sample gradient norm (via Hutch/Hutch++) to the true norm |
| $T$ | Trade-off function characterizing the $f$-DP guarantee of a single DP-SGD-RC step |
| $\sigma_Z$ | Noise multiplier scaled by $Z$, connecting randomized sensitivity to the Gaussian mechanism |
| $\preceq_{st}, \preceq_{cx}$ | Stochastic and convex ordering relations for the dominating privacy curve (Defs. A.13, A.14) |

*Table 5.* Key notation used throughout the paper.

## C. Ablation Studies

We conducted ablation studies on variation in the noise multiplier with projection dimension ($k$) and non-projected dimension of linear layer ($d$) for a given privacy parameter $\epsilon$ with the proposed method as compared to DP-SGD. Figure. 11b shows that the noise multiplier ($\sigma$) for a given $\epsilon$ increases with decrease in $k$ as expected. Smaller projection dimension results in larger error in norm estimation and therefore requires higher noise multiplier for the same privacy settings. However, for a projection dimension of 32 or higher and with a fixed $\epsilon$, the noise multiplier of the proposed method is similar to that of the baseline DP-SGD method. Further, Figure. 12 shows the variation in noise multiplier with respect to the non-projection dimension $d$ for $\epsilon = 2$ and 9. The noise multiplier for a given $\epsilon$ monotonically increases with $d$ and rapidly converges

by $d \sim 512$. The noise multipliers for Hutch$^{++}$ are larger than those of Hutch since the corresponding envelope CDF of Hutch$^{++}$ dominates that of Hutch. As $d$ increases, this gap reduces, and so does the gap in noise multipliers.

Table 6 reports the noise multiplier and accuracy on the BBC dataset for full fine-tuning with $\varepsilon = 2$ for varying projection dimensions $k \in \{8, 32, 64, 512\}$. We observe that as $k$ increases, the noise multiplier decreases and converges, with accuracy stabilizing at $95.4\%$ for $k \geq 64$, suggesting that moderate projection dimensions suffice for strong utility.

We also conducted an simulation study on the relative norm estimation error for different sized matrices. We report the relative norm estimation error in Table 7 (with 95% confidence intervals) for HUTCH and HUTCH$^{++}$ across random matrices of dimensions up to $16384 \times 16384$ with context length upto $T = 16384$, – this corresponds to the largest layers in frontier models such as Llama 4. The results empirically verify the relative norm estimation errors are independent of the dimension of the matrix and completely dependent on the projection dimension.

**Recommendation for Projection Dimension** Across all ablations, $k \in [32, 64]$ offers a strong balance between memory efficiency and utility. Increasing $k$ beyond this range yields negligible utility gains while incurring additional memory overhead. Moreover, standard stochastic trace estimation theory along with simulation results (Table 7) indicate the relative error depends only on $k$ and not on $d$, $p$, or $T$. Hence, this recommendation holds universally across model sizes.

| $k$ | Noise Multiplier | Accuracy |
|-----|------------------|----------|
| 8   | 3.1563           | 88.0     |
| 32  | 1.821            | 95.1     |
| 64  | 1.774            | 95.4     |
| 512 | 1.7568           | 95.4     |

*Table 6.* Utility ablation on BBC (full fine-tuning, $\varepsilon = 2$) varying $k$.

| $T$ | $d = p$ | Hutch | Hutch++ |
|-----|---------|-------|---------|
| 2048 | 2048 | 2.12e-01 $\pm$ 3.16e-02 | 1.35e-05 $\pm$ 1.91e-06 |
| 2048 | 4096 | 1.69e-01 $\pm$ 2.37e-02 | 1.25e-05 $\pm$ 1.86e-06 |
| 2048 | 8192 | 1.94e-01 $\pm$ 2.42e-02 | 1.01e-05 $\pm$ 1.37e-06 |
| 2048 | 16384 | 1.89e-01 $\pm$ 2.83e-02 | 9.82e-06 $\pm$ 1.56e-06 |
| 4096 | 2048 | 2.07e-01 $\pm$ 3.22e-02 | 6.23e-06 $\pm$ 1.07e-06 |
| 4096 | 4096 | 1.97e-01 $\pm$ 2.96e-02 | 5.78e-06 $\pm$ 8.40e-07 |
| 4096 | 8192 | 2.15e-01 $\pm$ 3.84e-02 | 5.50e-06 $\pm$ 7.99e-07 |
| 4096 | 16384 | 1.75e-01 $\pm$ 2.43e-02 | 4.34e-06 $\pm$ 5.79e-07 |
| 8192 | 2048 | 1.96e-01 $\pm$ 2.87e-02 | 3.21e-06 $\pm$ 4.54e-07 |
| 8192 | 4096 | 2.09e-01 $\pm$ 2.92e-02 | 7.55e-06 $\pm$ 4.61e-07 |
| 8192 | 8192 | 2.23e-01 $\pm$ 3.46e-02 | 7.95e-06 $\pm$ 4.44e-07 |
| 8192 | 16384 | 1.81e-01 $\pm$ 2.95e-02 | 7.04e-06 $\pm$ 3.51e-07 |
| 16384 | 2048 | 1.94e-01 $\pm$ 3.21e-02 | 3.87e-06 $\pm$ 3.86e-07 |
| 16384 | 4096 | 1.88e-01 $\pm$ 2.84e-02 | 2.89e-06 $\pm$ 2.44e-07 |
| 16384 | 8192 | 2.09e-01 $\pm$ 3.42e-02 | 2.60e-06 $\pm$ 2.05e-07 |
| 16384 | 16384 | 2.05e-01 $\pm$ 2.88e-02 | 8.69e-07 $\pm$ 1.23e-07 |

*Table 7.* Relative error ($\pm 95\%$ CI) for Hutch and Hutch$^{++}$ estimators across varying $T$ and $d = p$.

## D. Hyper-parameters

Hyperparameter tuning was performed consistently across all experiments. We used a batch size of 64 for the BBC experiments (for both full finetuning and LoRA), while a batch size of 256 was used for all other datasets across both private and non-private settings and for both LoRA and non-LoRA training. For all LoRA experiments, the LoRA rank was set to 16 with $\alpha = 8$ and a dropout rate of 0.05. We trained the BBC and Billsum datasets for 10 and 3 epochs, respectively, for both full finetuning and LoRA, while the HotpotQA dataset was trained for a single epoch under full finetuning. AdamW (Loshchilov & Hutter, 2017) was used for all non-private experiments, and Clipped + Noisy AdamW was used for all private experiments. Bayesian hyperparameter search was conducted using the training loss as the optimization signal; accuracies on

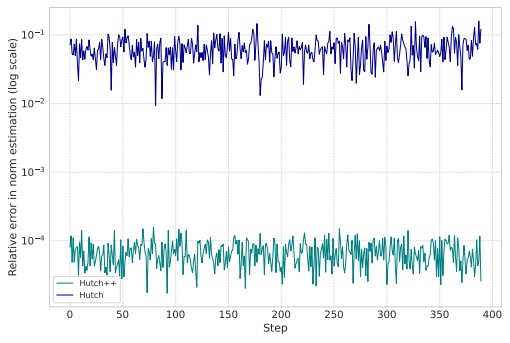

*(a)* Relative error in norm estimation

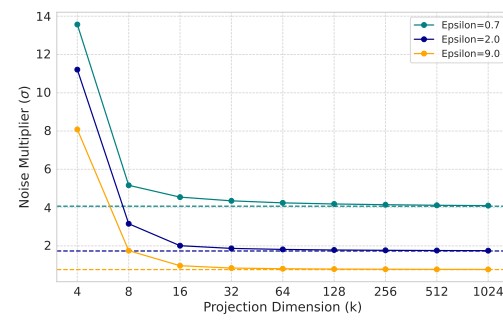

*(b)* Noise multiplier as a function of $k$.

*Figure 11.* (a) Relative error in norm estimation with Hutch and Hutch$^{++}$ for $(\epsilon : 0.7, \delta : 1e^{-5})$ for projection dimension ($k$) set to 32. (b) Noise multiplier as a function of projection dimension ($k$) for various epsilons for DP-SGD-RC with Hutch. Both plots correspond to the Full Fine-tuning experimental setup of BBC dataset.

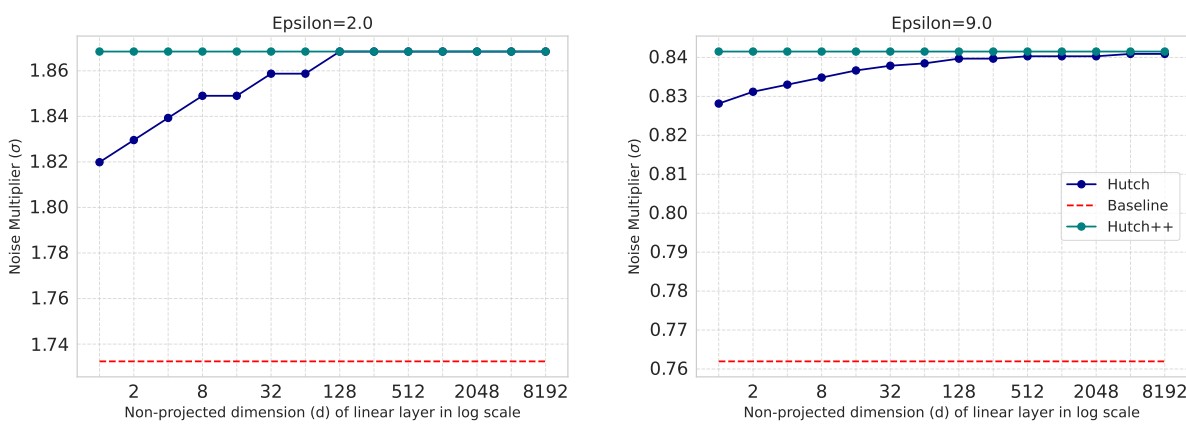

*Figure 12.* Noise multiplier ($\sigma$) as a function of hidden (non-projected) dimension ($d$) of the linear layer for a fixed projection dimension ($k$) of 32 for Hutch, Hutch$^{++}$ and DP-SGD (Baseline). We used BBC dataset experimental setup for both the plots.

the test set were then computed for the top three models and the best accuracy among them was reported. All hyperparameter values are reported in Table 8. It is important to note that for BBC, Billsum, and HotpotQA, the hyperparameters used for the privacy baseline at $\varepsilon = 2$ and for all randomized clipping experiments were reused from those obtained by tuning the baseline model with $\varepsilon = 9$ on the corresponding dataset. Hence, all the private experiments for a dataset in a single way of training have the same hyperparameters. For the full-finetuning experiments, we fine-tuned all the layers for the BBC and HotPotQA dataset while for the BillSum dataest, we only finetuned the last 5 layers. For LoRA experiments, we use a rank of 16 and fine-tune key, query and value layers for BBC dataset and all the linear layers in the attention block (including gate, up-proj, and down-proj layers) for BillSum dataset.

# E. Memory and Compute Analysis Details

This section provides a detailed memory and compute analysis comparing Fast Gradient Clipping (FGC), Ghost Clipping (GC), and DP-SGD-RC with the Hutchinson estimator. Table 9 summarizes the peak memory and memory overhead (defined as peak memory minus initial memory) for computing per-sample gradients in a linear layer without bias. We present the analysis both with and without deletion of output gradients (i.e., backprops). When the book-keeping algorithm (Bu et al., 2021) is not employed, backprops can be deleted during the first backward pass immediately after they are used for

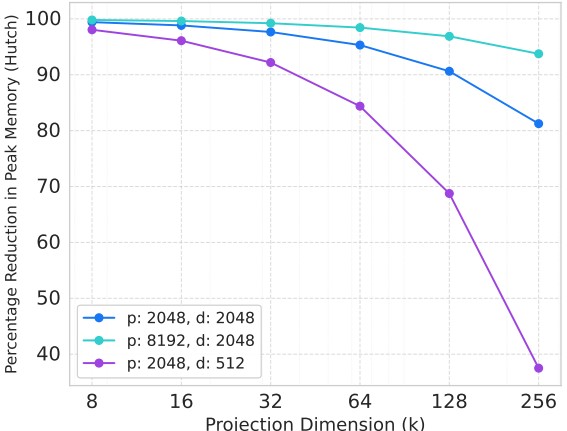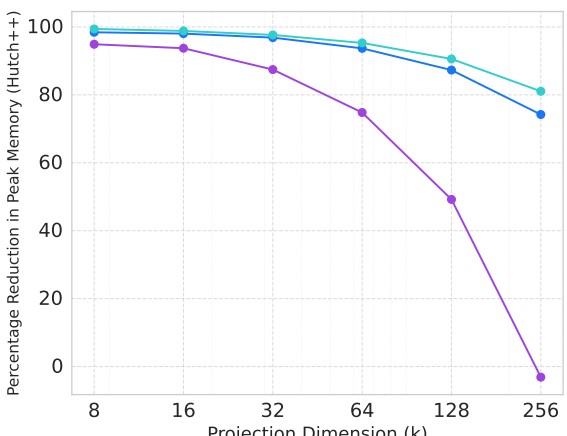

*Figure 13.* Peak memory savings for full fine-tuning settings without considering inputs (activations and backprops) memory as function of projection dimension for different linear layers of Llama3.2 1B.

per-sample norm computation, thereby reducing the memory footprint. For DP-SGD-RC, this optimization reduces the memory overhead from $Bk(T+d) + pk$ to $\max(BTk + pk, \ Bk(d-p) + Bdk)$. In the regime where $d < p < T$, the memory overhead of DP-SGD-RC simplifies to $BTk + pk$.

We now analyze the regimes in which DP-SGD-RC outperforms the mixed-ghost clipping baseline, which selects between FGC and GC based on $\min(pd, T^2)$. We assume that all methods delete backprops when they are no longer needed. Specifically, we seek to identify when the memory overhead of DP-SGD-RC, given by $\max(BTk + pk, \ Bk(d-p) + Bdk)$, is less than that of mixed-ghost clipping, given by $\min(Bpd, \max(BT^2, BT(2T-p)))$. For simplicity, we assume $p \geq d$. Table 10 presents the conditions on the projection dimension $k$ under various regimes for DP-SGD-RC to achieve the lowest memory footprint. Using these conditions, one can determine the valid range of $k$ for any combination of $p, d, T$, and $B$. The last column provides example ranges of valid context lengths $T$ for fixed $p, d$, and $B = 2$. We observe that DP-SGD-RC achieves the minimal memory footprint compared to both FGC and GC across a wide range of context lengths in all regimes.

Table 11 presents the FLOPs analysis for computing per-sample gradient norms in a linear layer. Fast Gradient Clipping (FGC) requires $O(BTpd)$ operations as it explicitly computes per-sample gradients through matrix multiplication of backprops and activations. Ghost Clipping (GC) avoids materializing gradients but incurs a $O(BT^2(p+d))$ cost due to the computation of Gram matrices $XX^\top$ and $\delta\delta^\top$, each of size $T \times T$. In contrast, DP-SGD-RC achieves $O(BTk(p+d))$ complexity by projecting backprops to a $k$-dimensional subspace before computing the norm estimate. Since $k \ll \min(T, pd/(p+d))$ in practice (e.g., $k = 32$ while $pd/(p+d) = 1024$ for $p = d = 2048$), DP-SGD-RC provides substantial computational savings over both GC and FGC. Specifically, DP-SGD-RC reduces FLOPs by a factor of $T/k$ compared to GC and by a factor of $pd/(k(p+d))$ compared to FGC. For example, with $T = 4096$, $p = d = 2048$, and $k = 32$, DP-SGD-RC reduces FLOPs by approximately $128\times$ compared to GC and $32\times$ compared to FGC, making it particularly well-suited for long-context language models.

| Dataset | Setting | Full Finetuning | LoRA |
|---|---|---|---|
| BBC | Non-Private | WD: 0.0001
LR: $\left[3.6 \times 10^{-5}, 3.7 \times 10^{-5}\right]$ | WD: 0.0001
LR: $\left[3.6 \times 10^{-5}, 3.7 \times 10^{-5}\right]$ |
| | Private | WD: $\left[3.8 \times 10^{-6}, 3.9 \times 10^{-6}\right]$
LR: 0.00025
C: 0.1 | WD: $\left[3.8 \times 10^{-6}, 3.9 \times 10^{-6}\right]$
LR: 0.00025
C: 0.1 |
| Billsum | Non-Private | WD: $\left[1.022 \times 10^{-4}, 1.023 \times 10^{-4}\right]$
LR: $\left[1.424 \times 10^{-4}, 1.425 \times 10^{-4}\right]$ | WD: $\left[5.18 \times 10^{-5}, 5.19 \times 10^{-5}\right]$
LR: $\left[4.83 \times 10^{-4}, 4.84 \times 10^{-4}\right]$ |
| | Private | WD: $\left[4.11 \times 10^{-2}, 4.12 \times 10^{-2}\right]$
LR: $\left[2.5 \times 10^{-5}, 2.6 \times 10^{-5}\right]$
C: $[0.366, 0.367]$ | WD: $\left[5.18 \times 10^{-5}, 5.19 \times 10^{-5}\right]$
LR: $\left[4.83 \times 10^{-4}, 4.84 \times 10^{-4}\right]$
C: $\left[10^{-2}, 1.1 \times 10^{-2}\right]$ |
| HotpotQA | Non-Private | WD: $\left[8 \times 10^{-6}, 9 \times 10^{-6}\right]$
LR: $\left[6 \times 10^{-6}, 7 \times 10^{-6}\right]$ | N.A. |
| | Private | WD: $\left[8 \times 10^{-6}, 9 \times 10^{-6}\right]$
LR: $\left[6 \times 10^{-6}, 7 \times 10^{-6}\right]$
C: $[0.005, 0.006]$ | N.A. |

*Table 8.* Hyperparameters for each dataset and training setting. WD: weight decay, LR: learning rate, C: clipping norm.

*Table 9.* Peak memory analysis for various DP-SGD based methods of a linear layer without bias. Note that "backprops" (aka output gradients) can be deleted when book-keeping based DP-SGD is not used. Initial memory for all methods: $BT(d + p)$, activations size: $B \times T \times d$, backprops size: $B \times T \times p$, projection dimension: $k$, and batch-size: $B$.

| Method | del backprops | Peak Memory | (Peak − Initial) Memory |
|---|---|---|---|
| FGC | × | $BT(d + p) + Bpd$ | $Bpd$ |
| FGC | ✓ | $BT(d + p) + Bpd$ | $Bpd$ |
| GC | × | $BT(d + p) + 2BT^2$ | $2BT^2$ |
| GC | ✓ | $\max(BT(d + p + T),\ BT(d + 2T))$ | $\max(BT^2,\ BT(2T - p))$ |
| DP-SGD-RC | × | $BT(d + p + k) + pk + Bdk$ | $Bk(T + d) + pk$ |
| DP-SGD-RC | ✓ | $\max(BT(d + p + k) + pk,\ BT(d + k) + Bdk)$ | $\max(BTk + pk,\ BT(k - p) + Bdk)$ |

*Table 10.* Conditions for DP-SGD-RC with Hutchinson estimator to have lowest peak memory assuming $p \geq d$.

| Regime | Conditions | DP-SGD-RC wins when | Valid $T$ for $B = 2$ and $k = 32$ |
|---|---|---|---|
| A-I | $p \geq 2d,\ T \leq p$ | $k < B \cdot \min(pd,\ T^2)\ /\ (BT + p)$ | $379 < T < 8192$ w/ $p = 8192, d = 2048$ |
| A-II | $p \geq 2d,\ T > p$ | $k < B \cdot \min(pd,\ T(2T - p))\ /\ (BT + p)$ | $8192 < T < 52$k w/ $p = 8192, d = 2048$ |
| B-I | $d \leq p < 2d,\ T \leq 2d - p$ | $k < B \cdot \min(pd,\ T^2)\ /\ (B(2d - p) + p)$ | $287 \leq T \leq 1024$ w/ $p = 3072, d = 2048$ |
| B-II | $d \leq p < 2d,\ 2d - p < T \leq p$ | $k < B \cdot \min(pd,\ T^2)\ /\ (BT + p)$ | $1024 < T \leq 3072$ w/ $p = 3072, d = 2048$ |
| B-III | $d \leq p < 2d,\ T > p$ | $k < B \cdot \min(pd,\ T(2T - p))\ /\ (BT + p)$ | $3072 < T < 195$k w/ $p = 3072, d = 2048$ |

*Table 11.* FLOPs analysis for various DP-SGD based methods of a linear layer without bias. Notation: activations size: $B \times T \times d$, backprops size: $B \times T \times p$, projection dimension: $k$, and batch-size: $B$.

| Method | Multiplications | Additions | Exact FLOPs | Order |
|---|---|---|---|---|
| FGC | $BTpd$ | $B(T-1)pd$ | $Bpd(2T - 1)$ | $O(BTpd)$ |
| GC | $BT^2(p + d + 1)$ | $BT^2(p + d - 1) - B$ | $2BT^2(p + d) - B$ | $O(BT^2(p + d))$ |
| DP-SGD-RC | $BTk(p + d) + Bdk$ | $BTk(p + d - 2) + Bdk - B$ | $2BTk(p + d) + Bk(d - T) - B$ | $O(BTk(p + d))$ |

