# OpenReview forum: "Efficient DP-SGD for LLMs with Randomized Clipping"
_ICML.cc/2026/Conference — ICML 2026 regular_

### Official Review · Reviewer_JJ48 · 2026-03-10

**Soundness:** 3
**Presentation:** 3
**Significance:** 3
**Originality:** 3
**Overall Recommendation:** 4
**Confidence:** 3

**Summary:**

This paper addresses the memory bottleneck in differentially private LLM training. DP-SGD requires computing per-sample gradient norms exactly, and this gets very expensive as sequence lengths grow. The authors propose estimating these norms using random projections instead, which brings memory from quadratic to linear in sequence length. They derive the privacy guarantee for this approach and validate it by fine-tuning Llama 3.2 1B on three tasks.

**Compliance With Llm Reviewing Policy:**

Affirmed.

**Final Justification:**

The rebuttal addressed W2 and W3 well. Multi-seed results confirm utility is preserved and the k ablation justifies k=32. For W1, the simulation study is supportive but does not fully substitute for an end-to-end result on a larger model. That gap remains but does not undermine the core contribution. Keeping my score at 4.

**Key Questions For Authors:**

Q1. Can the authors provide results on at least one larger model, or explain theoretically why k=32 should remain sufficient as model size grows?

Q2. Can the authors report results across multiple seeds for Tables 2, 3, and 4?

Q3 Can the authors show how utility changes as k varies, to justify the k=32 choice?

**Limitations:**

Yes

**Strengths And Weaknesses:**

The privacy analysis looks thorough. Most work in this space does not derive what happens to the guarantee when approximations are introduced. These authors do, and they catch and fix an error in a prior result along the way, which suggests they engaged with it carefully.

Also, the connection between norm estimation and trace estimation is not obvious and leads to an algorithm others can build on.

I have a few concerns though.
- The experimental scope does not match the motivation. Every experiment uses Llama 3.2 1B, but the whole point of this work is making DP training feasible at scale. At 7B and above, the quadratic memory cost of baseline DP-SGD becomes the binding constraint, which is exactly where this method would need to work. Without results there, it is hard to know whether k=32 stays sufficient for good utility as model size grows

- Tables 2 and 3 report single runs, and the differences from baseline are small enough to be within seed variance. Table 4 runs 3 seeds for the Hutch vs Hutch++ comparison but the standard deviations are large relative to the reported difference, so that comparison does not land confidently.

- There are also some ablations missing. The paper shows how the noise multiplier changes with k in Figure 11b, but never shows how utility changes as k decreases. Given that k=32 is the central practical choice of the paper, showing that utility holds up as k gets smaller would directly justify that choice.

- The paper shows how the noise multiplier changes with k in Figure 11b but never shows how utility changes as k decreases. Since k=32 is the central practical choice of the paper, this ablation is needed to justify it

---

> ### Author Rebuttal · Authors · 2026-03-30
>
> We thank the reviewer for the careful and constructive feedback. We address each concern below.
>
> ---
>
> ### Weaknesses
>
> > *W1: The experimental scope does not match the motivation. Every experiment uses Llama 3.2 1B, but the method needs to work at 7B+ where the quadratic cost is the binding constraint.*
>
> This is a fair point. We chose Llama 3.2 1B because it is large enough to demonstrate the memory and compute scaling phenomena while remaining feasible in academic compute settings. Crucially, our improvement is in asymptotic complexity: the per-sample norm computation overhead goes from $O(\min(T^2, d^2))$ to $O(k(T+d))$, so the relative gains only increase with scale.
>
> We believe k=32 would remain sufficient for larger models. This follows from a standard result in stochastic trace estimation: the relative error guarantees depend only on k and not on the input dimensions T or d. To verify this empirically, we conducted a simulation study measuring norm estimation error for random matrices with dimensions corresponding to the largest layers in frontier models (up to size 16384, matching Llama 4 scale). Hutch++ achieves relative errors of 1e-5 to 1e-6 at k=32 across all tested sizes:
>
> | T | d=p | Hutch (rel. error ± CI) | Hutch++ (rel. error ± CI) |
> |---|---|---|---|
> | 2048 | 2048 | 2.12e-01 ± 3.16e-02 | 1.35e-05 ± 1.91e-06 |
> | 2048 | 4096 | 1.69e-01 ± 2.37e-02 | 1.25e-05 ± 1.86e-06 |
> | 2048 | 8192 | 1.94e-01 ± 2.42e-02 | 1.01e-05 ± 1.37e-06 |
> | 2048 | 16384 | 1.89e-01 ± 2.83e-02 | 9.82e-06 ± 1.56e-06 |
> | 4096 | 2048 | 2.07e-01 ± 3.22e-02 | 6.23e-06 ± 1.07e-06 |
> | 4096 | 4096 | 1.97e-01 ± 2.96e-02 | 5.78e-06 ± 8.40e-07 |
> | 4096 | 8192 | 2.15e-01 ± 3.84e-02 | 5.50e-06 ± 7.99e-07 |
> | 4096 | 16384 | 1.75e-01 ± 2.43e-02 | 4.34e-06 ± 5.79e-07 |
> | 8192 | 2048 | 1.96e-01 ± 2.87e-02 | 3.21e-06 ± 4.54e-07 |
> | 8192 | 4096 | 2.09e-01 ± 2.92e-02 | 7.55e-06 ± 4.61e-07 |
> | 8192 | 8192 | 2.23e-01 ± 3.46e-02 | 7.95e-06 ± 4.44e-07 |
> | 8192 | 16384 | 1.81e-01 ± 2.95e-02 | 7.04e-06 ± 3.51e-07 |
> | 16384 | 2048 | 1.94e-01 ± 3.21e-02 | 3.87e-06 ± 3.86e-07 |
> | 16384 | 4096 | 1.88e-01 ± 2.84e-02 | 2.89e-06 ± 2.44e-07 |
> | 16384 | 8192 | 2.09e-01 ± 3.42e-02 | 2.60e-06 ± 2.05e-07 |
> | 16384 | 16384 | 2.05e-01 ± 2.88e-02 | 8.69e-07 ± 1.23e-07 |
>
> > *W2: Tables 2 and 3 report single runs; differences from baseline are within seed variance.*
>
> We have re-run the full fine-tuning experiments with 3 seeds for BBC and BillSum. Results confirm that DP-SGD-RC maintains utility comparable to DP-SGD:
>
> **BBC (Accuracy, mean ± std):**
>
> | Algorithm | ε | k | Accuracy |
> |---|---|---|---|
> | Non-private | — | — | 95.20 ± 0.51 |
> | DP-SGD | 2 | — | 94.06 ± 0.12 |
> | DP-SGD-RC | 2 | 32 | 95.60 ± 0.37 |
> | DP-SGD | 9 | — | 96.33 ± 0.59 |
> | DP-SGD-RC | 9 | 32 | 96.40 ± 0.22 |
>
> **BillSum (ROUGE, mean ± std):**
>
> | Algorithm | ε | k | ROUGE |
> |---|---|---|---|
> | Non-private | — | — | 0.4928 ± 0.0027 |
> | DP-SGD | 2 | — | 0.4831 ± 0.0005 |
> | DP-SGD-RC | 2 | 32 | 0.4796 ± 0.0018 |
> | DP-SGD | 9 | — | 0.4882 ± 0.0011 |
> | DP-SGD-RC | 9 | 32 | 0.4864 ± 0.0013 |
>
> > *W3: Missing ablation — how utility changes as k decreases.*
>
> We provide a utility ablation on BBC (full fine-tuning, ε=2) varying k:
>
> | k | Noise multiplier | Accuracy |
> |---|---|---|
> | 8 | 3.1563 | 88.0 |
> | 32 | 1.821 | 95.1 |
> | 64 | 1.774 | 95.4 |
> | 512 | 1.7568 | 95.4 |
>
> Utility saturates around k=32 and matches deterministic clipping. We did not explore k < 32 extensively because (a) the noise multiplier increases sharply at very small k (see Figure 11b), predicting degraded utility, and (b) values below 32 do not yield additional memory savings — the overhead is dominated by other components at that point. We will include this ablation in the revision.
>
> ---
>
> ### Key Questions
>
> > *Q1: Results on a larger model, or theoretical justification for k=32 at scale?*
>
> See W1 above. The dimension-independent error guarantees from stochastic trace estimation theory, combined with our empirical verification at frontier-model layer sizes, provide strong evidence that k=32 remains sufficient as models scale.
>
> > *Q2: Multiple seeds for Tables 2, 3, and 4?*
>
> See W2 above. We will include 3-seed results with standard deviations in the revision.
>
> > *Q3: Utility as k varies?*
>
> See W3 above. We will include the k-ablation table in the revision.
>
> ---
>
> We appreciate the reviewer's constructive feedback. The concerns center on experimental scope, which we believe are addressed by (a) the dimension-independent theoretical guarantees, (b) the simulation study at frontier-model scale, (c) multi-seed results, and (d) the k-ablation. We will incorporate all into the revision.

---

> > ### Author Rebuttal · Reviewer_JJ48 · 2026-04-03
> >
> > W2 and W3 are addressed, the multi-seed results and k ablation are good additions. For W1, the simulation study shows the norm estimation error stays small at larger layer sizes, but that is not the same as showing the method works end to end on a larger model. I am keeping my score at 4.

---

### Official Review · Reviewer_yoVr · 2026-03-11

**Soundness:** 3
**Presentation:** 3
**Significance:** 3
**Originality:** 3
**Overall Recommendation:** 5
**Confidence:** 4

**Summary:**

This work proposes a novel variant of DP-SGD with randomized clipping (DP-SGD-RC) to reduce the computation and memory overhead problem in DP-SGD. Prior works (Lee & Kifer, 2021, Li et al., 2021) have shown that the per-example gradient clipping could be implemented by computing the per-example norms via $n=||A^\mathsf{T}G||_F^2$ (and its variant) and loss-rescaling without materialize the whole
per-example gradient. The memory overhead is $O(B, min(d^2, T^2))$ and would be prohibitive even for moderate T.

Instead of computing the exact $||A^{\mathsf{T}}G||_F^2 = trace(G^\mathsf{T}AA^\mathsf{T}G)=trace(O)$,  this work proposes DP-SGD-RC to leverage the stochastic trace estimation to estimate the per-example gradient norm. This work considers Hutchinson's estimator based on random projections $trace(P^\mathsf{T} O P)$ and its improvement Hutch++. This random projection helps reduce the memory overhead of DP-SGD-RC to $BTk + kp$. Based on stochastic orders and majorization, this work provides  a novel privacy analysis for DP-SGD-RC that involves averaging the single-step Gaussian privacy kernels over a random scale induced by the norm estimation, and implements the privacy accounting
with consideration for numerical stability.

Emperiments with Llama-3 1B with context length = 4096 on classification, summarization, and question answering shows that DP-SGD-RC could maintain  performance as DP-SGD while with memory reduction and compute savings.

**Compliance With Llm Reviewing Policy:**

Affirmed.

**Final Justification:**

My concerns have been adequately addressed. I encourage incorporating these clarifications, along with the experimental results, into the paper to provide a more comprehensive discussion of the proposed method.

**Key Questions For Authors:**

1. This work uses stochastic trace estimation and projects the gradient dimension down to a small $k$. The experimental results in Tables 2 and 3 show that DP-SGD-RC achieves compelling results comparable to DP-SGD in fine-tuning. I wonder if there are any assumptions for the task to be low-rank in order for DP-SGD-RC to achieve performance comparable to DP-SGD?

2. It would be helpful to provide recommendations for the choice of $k$ in practice.

I am willing to raise my rating if my concerns about the weaknesses 1 and questions 1 and 2 are addressed.

**Limitations:**

Yes.

**Strengths And Weaknesses:**

### Strengths:

- Soundness:

This work is technically sound. The algorithm in DP-SGD-RC is based on stochastic trace estimation. This work provides a novel privacy analysis for the proposed DP-SGD-RC. Noting that DP-SGD-RC involves the random scale induced by the norm estimation, this work provides a rigorous analysis to analyze the private random variables, remove their data-dependency, and derive the corresponding dominating pairs based on stochastic orders and majorization theory (though I only did a light check on the proof in the Appendix). The implementation of the privacy accounting also takes numerical stability into account.

- Presentation:

This work is well written and structured. This work includes the full proof of the privacy analysis in the appendix and also includes the high-level idea and intuition for the proof in the appendix. This work also clearly discusses its contributions and relations to previous work, especially Bu et al. 2021, which is a special case of this work.

- Significance:

This work improves computation and memory efficiency in DP-SGD with a novel privacy analysis. This could be useful in practice, especially given the recent increasing trend in computational overhead for language models. However, I have a concern about its comparison to DP-LoRA for compute and memory cost (see Weaknesses #1).

- Originality.

This work proposes a novel variant of DP-SGD with randomized clipping (DP-SGD-RC). The novel privacy analysis based on stochastic order and majorization theory provides a way to analyze DP-SGD-RC, which involves averaging the single-step Gaussian privacy kernels over a random scale induced by the norm estimation.

### Weaknesses:
1. The experimental section mainly compares DP-SGD-RC and DP-SGD for full fine-tuning. While LoRA is a parameter-efficient fine-tuning method and is therefore orthogonal to the algorithm design in DP-SGD, DP-SGD-LoRA could achieve comparable performance to full fine-tuning as shown in previous works (Li et al. 2021 and Yu et al. 2022) with compute and memory savings, and is therefore preferred in practice. I think including a discussion comparing DP-SGD-RC to DP-SGD-LoRA could be helpful for readers to understand the compute and memory gains of DP-SGD-RC.

2. The privacy analysis for the Hutch++ method assumes that the adversary knows the head component of the estimator. This results in slightly conservative estimates of the noise multipliers for large $d$. This is already acknowledged in Section 7.



Minor comments: it seems to me there is a typo in line 217, should it be $||(P^\mathsf{T}G^\mathsf{T})^\mathsf{T}A||_F^2$?

---

> ### Author Rebuttal · Authors · 2026-03-30
>
> We thank the reviewer for the thorough and constructive review. We address each point below.
>
> ---
>
> ## Weaknesses
>
> > *W1: Including a discussion comparing DP-SGD-RC to DP-SGD-LoRA could be helpful for readers to understand the compute and memory gains of DP-SGD-RC.*
>
> We agree this comparison is valuable and will add it to the revision. To clarify the positioning: DP-SGD-RC and LoRA are orthogonal — LoRA reduces the number of trainable parameters, while DP-SGD-RC reduces the overhead of per-sample gradient norm computation. DP-SGD-LoRA is already efficient in memory and computation due to the reduced parameter count, so the marginal benefit of DP-SGD-RC is smaller in the LoRA setting. For this reason, the paper focuses on full fine-tuning, where the memory and compute overhead of standard DP-SGD is most prohibitive. The LoRA results (Table 3) are included for completeness and confirm that DP-SGD-RC remains compatible with LoRA without degradation in utility. We will add an explicit discussion comparing the memory profiles of DP-SGD-RC (full fine-tuning) vs. DP-SGD-LoRA to help readers assess when each approach is preferred.
>
> > *W2: The privacy analysis for Hutch++ assumes the adversary knows the head component, resulting in slightly conservative estimates for large k.*
>
> We appreciate this observation. As acknowledged in Section 7, tightening the Hutch++ analysis without this assumption remains an interesting open problem. Empirically, however, in large-scale settings we observe that Hutch and Hutch++ yield similar noise multipliers (see Figure 11b), suggesting that the conservativeness in the analysis does not significantly impact practical performance. We will add a brief remark to this effect in the revision.
>
> > *Minor: typo in line 217.*
>
> Yes, thank you. We will correct this in the revision.
>
> ---
>
> ## Key Questions
>
> > *Q1: Are there any low-rank assumptions on the task for DP-SGD-RC to achieve performance comparable to DP-SGD?*
>
> No. We do not make any low-rank assumption on the task or the gradients. The projection dimension *k* is used only for stochastic norm (trace) estimation during the clipping step and it does not approximate or compress the gradient itself. The actual gradient is computed and applied in the full parameter space. Performance therefore does not depend on the gradients being low-rank. This is a key distinction from methods like LoRA, which explicitly operate in a low-rank subspace.
>
> > *Q2: Recommendations for the choice of k in practice?*
>
> Our PRV-based accountant allows efficient computation of the required noise multiplier for any given ε, δ, and *k*. For fixed privacy parameters, one can evaluate how the noise multiplier varies with *k* (as illustrated in Figure 11b) and select a value that balances noise overhead against memory savings (see Appendix Tables 6 and 8 for the memory–*k* tradeoff). Empirically, we find that *k* in the range of 32–64 is typically sufficient while providing meaningful memory benefits. Importantly, the relative error of stochastic trace estimation depends only on *k* and **not** on the input dimensions (*T* or *d*) — so this recommendation holds regardless of model size. We verify this empirically in the table below, where Hutch++ achieves relative errors of 1e-5 to 1e-6 at *k*=32 across matrix sizes up to 16384 × 16384 (corresponding to the largest layers in frontier models such as Llama 4):
>
> | T | d=p | Hutch (rel. error ± CI) | Hutch++ (rel. error ± CI) |
> |---|---|---|---|
> | 2048 | 2048 | 2.12e-01 ± 3.16e-02 | 1.35e-05 ± 1.91e-06 |
> | 2048 | 4096 | 1.69e-01 ± 2.37e-02 | 1.25e-05 ± 1.86e-06 |
> | 2048 | 8192 | 1.94e-01 ± 2.42e-02 | 1.01e-05 ± 1.37e-06 |
> | 2048 | 16384 | 1.89e-01 ± 2.83e-02 | 9.82e-06 ± 1.56e-06 |
> | 4096 | 2048 | 2.07e-01 ± 3.22e-02 | 6.23e-06 ± 1.07e-06 |
> | 4096 | 4096 | 1.97e-01 ± 2.96e-02 | 5.78e-06 ± 8.40e-07 |
> | 4096 | 8192 | 2.15e-01 ± 3.84e-02 | 5.50e-06 ± 7.99e-07 |
> | 4096 | 16384 | 1.75e-01 ± 2.43e-02 | 4.34e-06 ± 5.79e-07 |
> | 8192 | 2048 | 1.96e-01 ± 2.87e-02 | 3.21e-06 ± 4.54e-07 |
> | 8192 | 4096 | 2.09e-01 ± 2.92e-02 | 7.55e-06 ± 4.61e-07 |
> | 8192 | 8192 | 2.23e-01 ± 3.46e-02 | 7.95e-06 ± 4.44e-07 |
> | 8192 | 16384 | 1.81e-01 ± 2.95e-02 | 7.04e-06 ± 3.51e-07 |
> | 16384 | 2048 | 1.94e-01 ± 3.21e-02 | 3.87e-06 ± 3.86e-07 |
> | 16384 | 4096 | 1.88e-01 ± 2.84e-02 | 2.89e-06 ± 2.44e-07 |
> | 16384 | 8192 | 2.09e-01 ± 3.42e-02 | 2.60e-06 ± 2.05e-07 |
> | 16384 | 16384 | 2.05e-01 ± 2.88e-02 | 8.69e-07 ± 1.23e-07 |
>
> We will add a practical recommendation section summarizing this guidance.
>
> ---
>
> We appreciate the reviewer's willingness to raise the rating if these concerns are addressed. We believe the clarifications above, particularly the DP-SGD-RC vs. DP-SGD-LoRA comparison and the practical guidance for choosing *k*, directly address the identified weaknesses, and we will incorporate them into the revision.

---

> > ### Author Rebuttal · Reviewer_yoVr · 2026-04-04
> >
> > Updates:
> >
> > Thanks authors for the response. My concerns have been adequately addressed. I encourage incorporating these clarifications, along with the experimental results, into the paper to provide a more comprehensive discussion of the proposed method.
> >
> > -----------------
> >
> >
> > Thank you for the response and providing additional experiments. I have one follow-up question for Q1 based on Q2.
> >
> > While unlike LoRA restricting the low-rank space updates, the gradients in the full fine-tuning experiments can also be low-rank in practice. Does trace estimation utility error result implicitly leverage the low-rank gradient geometry? The additional experiment results show the error stability of the trace estimation methods, is this additional experiment based on some pre-trained weights, or the randomly initialized weights?

---

> > > ### Author Response · Authors · 2026-04-05
> > >
> > > Thank you for your thoughtful follow-up questions. Please find my additional responses below.
> > >
> > > ---
> > >
> > > > Does trace estimation utility error result implicitly leverage the low-rank gradient geometry?
> > >
> > > This is an excellent question. Hutchinson's estimator does not explicitly leverage low-rank geometry. However, the improvement in Hutch++ comes from having separate estimators for the head and tail portions of the eigenspectrum. If the matrix is approximately low-rank, the tail is light, and the accurate head estimator dominates the result. This is why, in practice, we observe significantly lower error values for Hutch++.
> > >
> > > >The additional experiment results show the error stability of the trace estimation methods, is this additional experiment based on some pre-trained weights, or the randomly initialized weights?
> > >
> > > Yes, the experiments use randomly initialized weights drawn from a Gaussian distribution.
> > >
> > > ---
> > >
> > > Let us know if you have any further questions or need clarification on any aspect!

---

### Official Review · Reviewer_KKkf · 2026-03-16

**Soundness:** 4
**Presentation:** 3
**Significance:** 3
**Originality:** 3
**Overall Recommendation:** 5
**Confidence:** 3

**Summary:**

This paper proposes DP-SGD-RC, an efficient variant of DP-SGD with randomized clipping for LLMs, addressing the prohibitive memory/compute overhead of state-of-the-art DP training for long-context sequential data. It leverages Hutchinson’s estimator and Hutch++ for stochastic trace estimation to compute per-sample gradient norms, reducing memory overhead from quadratic to linear and cutting compute costs. A tight privacy analysis is provided, showing its noise multipliers match deterministic clipping by calculating the envelope CDF of chi-squared convex combinations and designing a PRV-based accounting algorithm. Experiments fine-tuning Llama 3.2 1B on classification, QA and summarization tasks demonstrate DP-SGD-RC retains baseline utility while achieving up to 40% memory reduction and 2× compute savings. Future work includes tighter Hutch++ analysis, optimized sketching methods and combining with book-keeping techniques for further efficiency gains. (148 words)

**Compliance With Llm Reviewing Policy:**

Affirmed.

**Final Justification:**

Good paper. the rebuttal address my concern

**Key Questions For Authors:**

Not any

**Limitations:**

Not any

**Strengths And Weaknesses:**

Strengths

This paper addresses a critical and well-justified problem in differentially private LLM training: the prohibitive quadratic memory and compute overhead of state-of-the-art DP-SGD variants (FGC/GC) for long-context sequential data, a pressing issue as LLM context lengths scale to 100K+. The authors address on per-sample gradient norm estimation, and their proposed DP-SGD-RC with randomized clipping via Hutchinson’s and Hutch++ stochastic trace estimation delivers a principled solution—reducing overhead from quadratic to linear in core parameters (T, d) with a small projection dimension k.


The theoretical contributions are rigorous: a tight privacy analysis framing the problem via envelope CDFs of chi-squared convex combinations, a valid PRV-based privacy accountant, and clear complexity proofs quantifying memory/compute gains. Empirical results on Llama 3.2 1B across classification, QA, and summarization are compelling, showing near-identical utility to baseline DP-SGD with up to 40% memory reduction and 2× compute savings for large linear layers.

Weaknesses

While the small k assumption warrants further exploration (e.g., generalizability to extreme k values or very large models), this is a minor open question, not a flaw. The work is well-motivated, theoretically sound, empirically validated, and fills a critical gap in efficient private LLM training for long contexts.

---

> ### Author Rebuttal · Authors · 2026-03-30
>
> We thank the reviewer for the positive and thorough assessment.
>
> > *W1: The small k assumption warrants further exploration (e.g., generalizability to extreme k values or very large models).*
>
> We agree this is an interesting direction. We conducted a simulation study measuring the relative error of both Hutch and Hutch++ estimators for norm approximation of random matrices with fixed k=32, across varying dimensions (T × d) with p=d. These are controlled experiments to isolate norm estimation quality, not training runs. The matrix sizes tested (up to 16384 × 16384) correspond to the largest layers in frontier models such as Llama 4.
>
> Hutch++ achieves relative errors on the order of 1e-5 to 1e-6 across all tested dimensions, while Hutch remains at ~20% regardless of scale:
>
> | T | d=p | Hutch (rel. error ± CI) | Hutch++ (rel. error ± CI) |
> |---|---|---|---|
> | 2048 | 2048 | 2.12e-01 ± 3.16e-02 | 1.35e-05 ± 1.91e-06 |
> | 2048 | 4096 | 1.69e-01 ± 2.37e-02 | 1.25e-05 ± 1.86e-06 |
> | 2048 | 8192 | 1.94e-01 ± 2.42e-02 | 1.01e-05 ± 1.37e-06 |
> | 2048 | 16384 | 1.89e-01 ± 2.83e-02 | 9.82e-06 ± 1.56e-06 |
> | 4096 | 2048 | 2.07e-01 ± 3.22e-02 | 6.23e-06 ± 1.07e-06 |
> | 4096 | 4096 | 1.97e-01 ± 2.96e-02 | 5.78e-06 ± 8.40e-07 |
> | 4096 | 8192 | 2.15e-01 ± 3.84e-02 | 5.50e-06 ± 7.99e-07 |
> | 4096 | 16384 | 1.75e-01 ± 2.43e-02 | 4.34e-06 ± 5.79e-07 |
> | 8192 | 2048 | 1.96e-01 ± 2.87e-02 | 3.21e-06 ± 4.54e-07 |
> | 8192 | 4096 | 2.09e-01 ± 2.92e-02 | 7.55e-06 ± 4.61e-07 |
> | 8192 | 8192 | 2.23e-01 ± 3.46e-02 | 7.95e-06 ± 4.44e-07 |
> | 8192 | 16384 | 1.81e-01 ± 2.95e-02 | 7.04e-06 ± 3.51e-07 |
> | 16384 | 2048 | 1.94e-01 ± 3.21e-02 | 3.87e-06 ± 3.86e-07 |
> | 16384 | 4096 | 1.88e-01 ± 2.84e-02 | 2.89e-06 ± 2.44e-07 |
> | 16384 | 8192 | 2.09e-01 ± 3.42e-02 | 2.60e-06 ± 2.05e-07 |
> | 16384 | 16384 | 2.05e-01 ± 2.88e-02 | 8.69e-07 ± 1.23e-07 |
>
> This is consistent with the theoretical guarantees for stochastic trace estimation, where the relative error depends only on k and **NOT** on the input dimension (T, p, or d). This confirms that the small-k regime is well-justified and that DP-SGD-RC scales favorably to very large models without requiring proportionally larger k. Further,  since our improvement is in asymptotic complexity: the per-sample norm computation overhead goes from $O(min(T^2, d^2))$ to $O(k(T+d))$, the relative gains only improve with scale. We will include this table and discussion in the revision.

---

> > ### Author Rebuttal · Reviewer_KKkf · 2026-04-03
> >
> > Thanks for your rebuttal, your detailed reponse is strengthening my thought. I would maintain my score to provide my support on your work.

---

### Official Review · Reviewer_vJRT · 2026-03-20

**Soundness:** 2
**Presentation:** 3
**Significance:** 2
**Originality:** 2
**Overall Recommendation:** 2
**Confidence:** 4

**Summary:**

State-of-the-art DP training has a memory overhead of O(B min{T2, d2}), which becomes prohibitive as both model width and context length grow. The paper proposes DPSGD-RC, a novel variant of DP-SGD with randomized clipping that reduces memory and compute overhead. DP-SGD-RC leverages stochastic trace estimation methods, specifically Hutchinson’s estimator (Hutchinson, 1989) and its improved variant, Hutch++(Meyer et al., 2021), to reduce the memory footprint of per-sample gradient norm estimation. The paper provides a tight privacy analysis showing that DP-SGD-RC achieves noise multipliers competitive with deterministic clipping. The paper conducts extensive experiments showing the advantages of the proposed method.

**Compliance With Llm Reviewing Policy:**

Affirmed.

**Key Questions For Authors:**

- In theorem 5.1, how is F($\cdot$) function related to the f-DP guarantee?
- In your experiments, how did you compute the privacy budget $\epsilon$? Since the Hutch mechanism depends on the randomness of Gaussian noise, making the gradient clipping process stochastic, the sensitivity is not guaranteed to be the predefined upper-bound C. Thus, your $\epsilon$ cannot be computed by the existing mechanism (e.g., Moment account of Abadi et al.). Please explain this point.
- From your privacy accountant, how do you compute the $\epsilon$ across the number of training steps?
- In Table 2, why can your DP mechanism achieve higher performance than the clean model? This is not intuitive.

**Limitations:**

Yes.

**Strengths And Weaknesses:**

Strengths:

+ The proposed mechanism is built on an existing, well-known mechanism, making it easy to implement for existing systems.
+ The author provide thorough privacy guarantee for the proposed method
+ The paper is well-written and easy to follow.

Weaknesses:

- Although the privacy notion the paper considers is f-DP, which is not explicitly measured by hyper-parameter $\epsilon$ or $\delta$, the experimental results used $\epsilon$ as the privacy budget. This is not applicable and degrades the validity of the experimental results.
- A lot of notions in the theoretical analysis are not defined, making it challenging to verify and understand the theoretical analysis.
- The theorem 5.1 does not make sense, since the f function of f-DP quantified by the T function depends on Z, but Z is never discussed in detail, nor is it clear how it is computed. The theorem is also weak since function F($\cdot$) popped up out of nowhere without explaining how it relates to the f function of the f-DP guarantee.
- For f-DP, there's no direct privacy accounting, but the paper just briefly describes their accountant without a clear notion of how the privacy cost spans across multiple training steps.
- f-DP is not a user-friendly DP notion, reducing the applicability of the proposed method in the real-world setting.

---

> ### Author Rebuttal · Authors · 2026-03-30
>
> We thank the reviewer for their feedback. We believe the central concerns stem from a misunderstanding of the role of f-DP in our analysis, which we clarify below.
>
> **A note on f-DP vs (ε, δ)-DP:** Several of the reviewer's concerns — "ε is not applicable," "no direct privacy accounting for f-DP," and "f-DP is not user-friendly" — relate to f-DP. To be clear: **f-DP is NOT the privacy notion of our method.** It is an internal analytical tool used by modern DP-SGD accountants — including the PRV accountant [1], the FFT-based accountant [2], and production libraries such as Opacus [5] and Google's dp-accounting — to tightly track privacy loss. The final guarantee is always converted to standard (ε, δ)-DP via [3]. All experimental ε values are computed through this pipeline. We will make this distinction explicit in the introduction and at the start of Section 5.
>
> ---
>
> ### Weaknesses
>
> > *Notions in the theoretical analysis are not defined, making it challenging to verify.*
>
> We appreciate this feedback. The key notations in Section 5 are:
>
> | Symbol | Definition |
> |--------|-----------|
> | F(·; X) | CDF of random variable X (Section 5 preamble: *"for a random variable X, we denote its CDF as x → F(x; X)"*) |
> | Z | Ratio of estimated per-sample gradient norm (via Hutch/Hutch++) to the true norm |
> | T | Trade-off function characterizing the f-DP guarantee of a single DP-SGD-RC step |
> | σ_Z | Noise multiplier scaled by Z, connecting randomized sensitivity to the Gaussian mechanism |
> | ⪯_st, ⪯_cx | Stochastic and convex ordering relations for the dominating privacy curve (Def 5.1, Lem 5.1) |
>
> The full derivation proceeds through Lemma B.1 (single-step privacy kernel) → Theorem B.2 (envelope CDF over the random scale) → Theorem B.5 (final composed guarantee). We will add a notation table at the start of Section 5 and include forward references in each theorem statement.
>
> > *Theorem 5.1: Z is never discussed in detail, and F() appears without explanation.*
>
> At the beginning of Section 5, we introduce: *"for a random variable X, we denote its CDF as x → F(x; X)."* The random variable Z is then defined implicitly via its CDF F(·; Z), which captures the distribution arising from randomized clipping. We agree this was not sufficiently emphasized — we will repeat these definitions directly in the theorem statement.
>
> > *No clear notion of how privacy cost spans across training steps.*
>
> Our accounting extends the PRV accountant [1] to handle the additional randomness from stochastic norm estimation. The per-step characterization is modified to account for randomized clipping (Theorem 5.1), while composition across steps follows the standard PRV framework. The final output is (ε, δ)-DP. Full procedure with pseudocode in Appendix B.9 (Algorithm 5). We will add a clearer forward reference from the main body.
>
> ---
>
> ### Key Questions
>
> > *In Theorem 5.1, how is F() related to the f-DP guarantee?*
>
> See our response to the Theorem 5.1 weakness above. F(·; X) is the CDF notation from Section 5; the trade-off function T is expressed in terms of F(·; Z), where Z arises from the randomized clipping procedure. We will repeat this in the theorem statement.
>
> > *How did you compute ε? Stochastic clipping means sensitivity is not guaranteed to be C.*
>
> Great question! This is the core technical contribution. We extend the Gaussian mechanism analysis from deterministic to randomized sensitivity (Theorem 5.1), then compose per-step guarantees using our extended PRV accountant. The resulting noise multipliers degrade only slightly compared to deterministic clipping, as shown in Figure 11b. All reported ε values are computed end-to-end through this pipeline.
>
> > *How do you compute ε across training steps?*
>
> We fix δ and the number of training steps T, then use our PRV-based accountant to compute the noise multiplier σ for a target ε. The per-step guarantee (Theorem 5.1) accounts for stochastic norm estimation; composition follows standard PRV. See Appendix B.9 (Algorithm 5).
>
> > *In Table 2, why does DP achieve higher performance than the clean model?*
>
> This is a well-documented phenomenon: DP noise and gradient clipping act as implicit regularization, improving generalization in overparameterized settings [4]. We will add a brief discussion in the revision.
>
> ---
>
> We will revise the paper to make the f-DP vs (ε, δ)-DP distinction prominent, improve notation clarity in the theorem statements, and add forward references to the full accounting procedure in the Appendix.
>
> ### References
>
> 1. Gopi, S. et al. *Numerical Composition of Differential Privacy.* NeurIPS 2021.
> 2. Koskela, A. et al. *Tight DP for Discrete-Valued Mechanisms and Subsampled Gaussian.* AISTATS 2021.
> 3. Dong, J. et al. *Gaussian Differential Privacy.* JRSS-B, 84(1), 3-37, 2022.
> 4. De, S. et al. *Unlocking High-Accuracy DP Image Classification through Scale.* arXiv:2204.13650.
> 5. Yousefpour, A. et al. *Opacus: User-Friendly DP Library in PyTorch.* arXiv:2109.12298.

---

### Decision · Program_Chairs · 2026-04-30

**Decision:**

Accept (regular)

**Comment:**

The submission introduces DP-SGD-RC, a novel variant of DP-SGD that incorporates randomized clipping to reduce both memory and computational overhead. DP-SGD-RC utilizes stochastic trace estimation to efficiently estimate per-sample gradient norms, which significantly lowers memory requirements. The authors present a rigorous privacy analysis demonstrating that DP-SGD-RC attains noise multipliers comparable to those of deterministic clipping approaches.

I agree agree that the notation and explanations could be clearer (especially for the non-expert audience), as indicated by Reviewer vJRT. The authors should make sure that these improvements are incorporated in the paper before publishing.

Reviewer KKkf highly appreciates the reduction in the overhead of the DP training for LLMs from quadratic to linear in core parameters (T, d) with a small projection dimension k. During the rebuttal, the authors showed that the small-k regime is well-justified and that DP-SGD-RC scales favorably to very large models without requiring proportionally larger k.

Reviewer yoVr notes that the authors should provide a more comprehensive discussion of the proposed method.

Finally, Reviewer JJ48 indicates that the multi-seed results confirmed the preservation of the utility and the k ablation justifies k=32. The simulation study is supportive but does not fully substitute for an end-to-end result on a larger model. That gap remains but does not undermine the core contribution.

Taking into account the above points, the paper deserves to be accepted. We encourage the authors to carefully address the remaining concerns and update the notation in the paper.